# Integration of tree hydraulic processes and functional impairment to capture the drought resilience of a semi-arid pine forest

Daniel Nadal-Sala[1,2], Rüdiger Grote[1,*], David Kraus[1], Uri Hochberg[3], Tamir Klein[4], Yael Wagner[4], Fedor Tatarinov[5], Dan Yakir[5], Nadine K. Ruehr[1,6]

[1]Institute of Meteorology and Climate Research (IMK-IFU), KIT-Campus Alpin, Karlsruhe Institute of Technology (KIT), Kreuzeckbahnstr. 19, 82467 Garmisch-Partenkirchen, Germany
[2]Centre de Recerca Ecològica i Aplicacions Forestals (CREAF), Campus de Bellaterra (UAB) Edifici C, 08193, Cerdanyola del Vallès, Spain
[3]Institute of Soil, Water and Environmental Sciences, Volcani Center, ARO, Rishon Lezion 7505101, Israel;
[4]Department of Plant Environmental Sciences, Weizmann Institute of Science, Rehovot 7610001, Israel
[5]Department of Earth and Planetary Sciences, Weizmann Institute of Science, Rehovot 76100, Israel
[6]Institute of Geography and Geoecology, Karlsruhe Institute of Technology (KIT), Karlsruhe 76131, Germany

*Correspondence to*: Rüdiger Grote (ruediger.grote@kit.edu)

**Abstract.** Drought stress imposes multiple feedback responses in plants. These responses span from stomata closure and enzymatic downregulation of photosynthetic activity to structural adjustments of xylem biomass and leaf area. Some of these processes are not easily reversible and may persist long after the stress ended. Despite a multitude of hydraulic model approaches, simulation models still widely lack an integrative mechanistic description on how this sequence of tree physiological to structural responses may be realized, which is also simple enough to be generally applicable.

Here, we suggest an integrative, sequential approach to simulate drought stress responses. Firstly, a decreasing plant water potential triggers stomatal closure alongside a downregulation of photosynthetic performance and thus effectively slows down further desiccation. A second protective mechanism is introduced by increasing the soil-root resistance, represented by a disconnection of fine roots after a threshold soil water potential has been reached. Further decreases in plant water potential due to residual transpiration and loss of internal stem water storage are consistently leading to a loss in hydraulic functioning reflected in sapwood loss and foliage senescence. This new model functionality has been used to investigate responses of tree hydraulics, carbon uptake and transpiration to soil- and atmospheric drought in an extremely dry Aleppo pine (*Pinus halepensis* L.) plantation.

Using the hypothesis of a sequential triggering of stress-mitigating responses, the model was able to reflect carbon uptake and transpiration patterns under varying soil water supply and atmospheric demand - especially during summer - and responded 30 realistically regarding medium-term responses such as leaf and sapwood senescence. We could show that the observed avoidance strategy was only achieved when the model accounted for a very early photosynthesis down-regulation, and the relatively high measured plant water potentials were well reproduced with a root-to-soil disconnection strategy that started

before major xylem conductance losses occurred. Residual canopy conductance was found to be pivotal in explaining dehydration and transpiration patterns during summer but it also disclosed that explaining the water balance in the driest

periods requires water supply from stem water and deep soil layers. In agreement with the high drought resistance observed at the site, our model indicated little loss of hydraulic functioning in Aleppo pine, despite the intensive seasonal summer drought.

## 1 Introduction

Reduced tree growth and increased tree mortality following hot and dry spells have been widely observed (e.g. Thom et al.,

2023; Ryan, 2011; Hammond et al., 2021). This trend is expected to extend into the future since rising vapor pressure deficit (VPD) and more irregular precipitation patterns are expected, leading to increases in drought severity (Huber et al., 2021; Tschumi et al., 2022). The extend of tree decline, however, also depends on the ability of tree species to withstand or respond to the stress. This includes responses that are not easily reversible after rewetting, and will therefore impact tree carbon and water balance also beyond stress, introducing so-called legacy effects (Ruehr et al., 2019).

To evaluate tree and forest responses to environmental changes, physiologically-oriented simulation models are essential tools because they are describing various physiological processes in dependence on environmental driving forces, which are then used to derive changes in biomass and dimensional growth (Fontes et al., 2010; Trugman et al., 2019). Nevertheless, the various known internal feedback responses to primary damages (López et al., 2021; Blackman et al., 2023), are still lacking a unified mechanistic formulation. A main reason for this deficit is that several tree processes are involved occurring at various

temporal scales. For example, stomatal closure may occur immediately and is easily reversible, but less reversible responses such as loss of xylem, roots or foliage are usually only observed after prolonged and/or severe drought stress (Barbeta and Peñuelas, 2016; Nadal-Sala et al., 2021a; Nardini et al., 2016). Immediate and intermediate responses during drought are stomatal closure, photosynthetic enzyme degradation and a decrease of mesophyll conductance (both regarded as non-stomatal limitations to photosynthesis) (Salmon et al., 2020; Dewar et al., 2018). This is followed by fine root retraction from the soil

which prevents potential water losses into the soil but also restricts further water uptake (Yang et al., 2023), and xylem embolism, which reduces conductance further but can lead to tree mortality (Brodribb and Cochard, 2009; Ruffault et al., 2022). Before death, however, the remaining evaporative demand can be further reduced by decreasing the evaporative surface itself, i.e. by shedding foliage (Blackman et al., 2023; Cardoso et al., 2020). These responses might be consistently considered by hierarchical triggering in dependence on each other, or by a sequential initiation at decreasing levels of plant hydration, for

example represented by plant water potential (Walthert et al., 2021) or plant water storage (Paschalis et al., 2024).

Current developments in physiologically-oriented stand level simulation models propose the calculation of hydraulic and stomatal conductance in dependence on leaf water potential (Kennedy et al., 2019; Christoffersen et al., 2016; Eller et al., 2018), which then drive losses in xylem conductance and leaf shedding (Xu et al., 2016), triggering tree mortality when drought

stress intensifies (Yao et al., 2022; Torres-Ruiz et al., 2024). Including non-stomatal limitations on photosynthesis based on leaf water potential has been first introduced by Tuzet et al. (2003). Afterwards, it has been identified as a key explanatory process for leaf exchange dynamics under sustained drought stress (Keenan et al., 2010; Yang et al., 2019; Gourlez de la Motte et al., 2020). In connection with a plant hydraulic model, it has been shown to result in more realistic water potential developments (Sabot et al., 2022). Still, it is difficult to realistically reproduce plant water potential developments while not sacrificing parsimony (Drake et al., 2017; Cochard et al., 2021). Despite the struggle on a simple but consistent and generally applicable solution, the need for a representation of hydraulic processes that also accounts for non-stomatal impacts is increasingly recognized. For example, drought stress realization can trigger the adjustment of allometric relations such as larger root-to-shoot and root-to-leaf ratios which are favouring water uptake and reducing water losses (Brunner et al., 2015; Lemaire et al., 2021), or competition processes at the stand level that lead to density-dependent tree mortality (Trugman, 2022; Pretzsch and Grote, 2024).

The full effect of such secondary responses can only be evaluated if not only the mitigating impact on tree water loss but also the trade-off in carbon acquisition and allocation change will be considered (Ruehr et al., 2019; Müller and Bahn, 2022). Degraded enzymes do need some days for recovery during which the full photosynthetic capacity is not available, and foliage that is lost during a drought event will slow-down desiccation allowing trees to survive longer under hydric stress (Blackman et al., 2023; Blackman et al., 2019) but will not be available for carbon assimilation after stress release, hindering recovery (Galiano et al., 2011). Also, impaired conducting tissue which is inherently related to potentially reduced carbon uptake, does hardly reverse quickly via refilling of embolism (also an energy-consuming process), but mostly depends on re-growth of new xylem tissue (Hammond et al., 2019; Rehschuh et al., 2020; Gauthey et al., 2022). Such secondary effects don't need a new functionality in ecosystem models but can be considered in existing integrated modelling frameworks, allowing the simulation of stress legacies in ecosystem process-based models. Moreover, the simulation of a mechanistic stress-driven tree mortality might be facilitated, if tissue function has been damaged beyond critical levels (McDowell et al., 2022; Breshears et al., 2018), or if regrowth and repair decrease resources for growing assimilating tissues with detrimental impacts on the acquisition of new carbon and nutrients (Bigler et al., 2007; McDowell et al., 2008; Rukh et al., 2023).

Previous attempts in incorporating explicit definitions on plant hydraulics in process-based models have been proven to capture instantaneous responses of leaf gas exchange to drought stress (e.g. De Kauwe et al., 2015a; Sperry et al., 2017; Tuzet et al., 2017; Cochard, 2021; Sabot et al., 2022). Similarly, plant hydraulics has also been used to investigate tree structural adjustment in response to drought stress, e.g. loss of xylem conductance due to cavitation (Whitehead et al., 1984; Tyree and Sperry, 1989), leaf shedding (Nadal-Sala et al., 2021a) and fine root biomass adjustments (Sperry et al., 1998). Overall, applying hydrological model schemes has been found promising to investigate plant strategies to minimize drought stress that are based on different trait expressions (Mirfenderesgi et al., 2019). However, modelling tree hydraulic processes at stand level is still challenging due to the complex interaction of environmental boundary conditions such as evaporative demand and soil properties, plant morphology (root distribution, individual size), anatomy (xylem conductivity and its vulnerability to

embolism), physiology (photosynthetic capacity, stomatal responsiveness) (Trugman et al., 2019; Mencuccini et al., 2019), and the representation of competition when shifting from single-tree to the stand level (Trugman, 2022). Specific problems are for example the consideration of tree capacitance (Blackman et al., 2019; Preisler et al., 2022), water loss after full stomatal closure (Barnard and Bauerle, 2013; Duursma et al., 2019), seasonal acclimation of xylem properties to low water potentials (Feng et al., 2023), or the issue of embolism recovery (Arend et al., 2022).

The importance to consider first level responses for drought stress mitigation as well as their trade-offs have been theoretically highlighted (Li et al., 2022; Trugman et al., 2019) and empirically demonstrated (e.g. Arend et al., 2022) but consistent model implementations are still scarcely found. Current approaches are either concentrating on instantaneous stomatal responses alone (Eller et al., 2020) or directly affecting tree mortality (Yao et al., 2022). The few physiologically-based approaches are computationally demanding and difficult to combine with established stand-level forest models (Ruffault et al., 2022), while the parameters required for an in-depth representations of the whole plant hydraulic pathway are manifold and difficult to calibrate against measurements. In order to investigate the implications of sequential hydraulic stress responses, we thus integrated tree hydraulic and stress impairment processes into an existing modelling framework, LandscapeDNDC (Haas et al., 2013). The approach is inspired by recent model innovations (e.g. De Kauwe et al., 2020; De Cáceres et al., 2021; Ruffault et al., 2022) but is not aiming at simulating precise soil and plant water potentials, because these depend on very specific soil and plant properties that are spatially heterogenic and highly dynamic. Instead, we propose a relatively simple but robust model scheme where soil water potentials are derived from generally available soil texture information and one average canopy water potential is assumed to impact responses of all leaves as well as the xylem (see Fig.1). The approach presented here, which is based on two simple but well established hydraulic principles and allometric relationships, is also representing major legacy mechanisms and medium-term feedbacks currently discussed (Trugman, 2022).

To evaluate this new model approach, we looked for a site that already exhibits the whole range of water availability reaching from none to very severe drought and at which long-term measurements exist able to constrain and evaluate the model processes. We have therefore focused on a seasonal-dry forest site dominated by Aleppo pine trees (*Pinus halepensis* Mill.) in Yatir, Israel. The site is characterized by a semi-arid climate with a short wet season in winter, followed by a prolonged dry summer period with no rain and high VPD (Wang et al., 2020). Considering that the forest might be at the verge of survival already while climate projections suggest an additional decrease of precipitation up to a 20%, (IPCC, 2019; D'Andrea et al., 2020) investigations that target the resilience of the trees might be of particular interest.

Our central physiological hypothesis states that a cascade of mechanisms in plants are triggered in response to declining water potential to prevent further dehydration (Novick et al., 2022). The intensity of such responses increases with decreasing water potential (Walthert et al., 2021), and the sensitivity of these responses is inversely related to the carbon costs of their reversal. The suggested model scheme (Fig. 1) represents a consistent implementation of this hypothesis, which will be tested by investigating the transition from wet to extreme dry conditions. In particular, we will target the following objectives: i) to evaluate the newly developed plant hydraulics module at an extreme seasonal dry forest site. In particular, the module is

challenged to represent the two main seasonal trends in Yatir regarding stomatal behaviour: VPD-driven stomatal limitation during times of ample soil moisture and soil moisture-driven limitations under dry environmental conditions. ii) to determine the potential importance of hydraulic-driven non-stomatal limitations on photosynthetic assimilation; and iii) to assess the impact of considering a root-to-soil disconnection process under realistic conditions of prolonged drought stress. Furthermore, we depict and discuss how the proposed hydraulic modelling scheme could be used to alter simulated leaf and sapwood area dynamics.

## 2 Materials and Methods

### 2.1 Site description

The Yatir forest (Israel, 31.34°N, 35.05°E) is an afforestation of Aleppo pine planted during the 1960's. The site conditions are characterized by an exceptionally dry climate with an annual precipitation of 285 mm, while potential evaporation is > 5-fold higher (Schiller, 2011; Ungar et al., 2013). Typically, the forest experiences a 6-8 month-long rain-free period during summer.

The soil at the site is a Rendzic Leptozol with an extremely clay-enriched layer at ca. 1 m depth, a permanent wilting point of 10.7% volumetric soil water content (SWC), and a high stone content (Klein et al., 2014; Preisler et al., 2019). Van Genuchten parameters have been directly derived from soil water retention curves, measured at four different depths at the investigation site (Klein et al., 2014). The threshold for water uptake has been set by a threshold parameter ($\Psi_{disconnect}$) which is calibrated to gas exchange (see section 2.3.3). This is, however, close to the water potential that develops at wilting point according to the initialized Van Genuchten parameters and the measured clay and sand content. The soil water potential at the wilting point decreased with depth and was slightly below -2 MPa in the upper 20 cm. During the study period, stand density was determined to be 357 trees ha$^{-1}$, average diameter at breast height of all trees was about 18.5 cm, and average tree height 9.3 m (based on Rohatyn, 2017 and personal communication). Natural regeneration is negligible (Pozner et al., 2022). Specific initializations for model simulations are given in the supplementary Table S1.

### 2.2 Observational data

Carbon and water fluxes and supplementary meteorological data are measured at a 19 m high flux tower in the geographical centre of the Yatir forest at the site. Weather variable include incoming photosynthetic active radiation, air temperature, vapor pressure deficit, wind speed and precipitation which are continuously recorded since the year 2000 (Grünzweig et al., 2003). Measurements are carried out according Euroflux standards and data are included in the CarboEuroFlux network (Aubinet et al., 1999). We selected the period between 2012 - 2015 for our study as it provides ample high-quality EC data as well as sap flux measurements and is freely available from the ICOS data portal (Warm Winter 2020 Team, 2022, https://www.icos-

cp.eu/data-products/2G60-ZHAK, visited 25.08.2023). We purposefully concentrated on a couple of years in order to omit any

potential impact from stand structural changes or increasing atmospheric $CO_2$ concentration (e.g. Norby et al., 2005).

EC measurements of net ecosystem production (NEP) and calculations of gross primary production (GPP) and ecosystem respiration (ER) using site-specific relations to temperature as described in Tatarinov et al. (2016) are provided at the ICOS data portal (Warm Winter 2020 Team, 2022, https://www.icos-cp.eu/data-products/2G60-ZHAK, visited 25.08.2023). Daily values were only calculated with good and very good NEP data quality, according to the Euroflux methodology. All other data

was considered as missing values. Days with > 2 half-hourly day-time values missing were excluded from the model evaluation (ca. 35%).

Sap flow measurements are based on up to sixteen trees using lab-manufactured thermal dissipation sensors (Granier and Loustau, 1994) at 30 min intervals. Sap flow was calculated following Granier and Loustau (1994), implementing corrections (Kanety et al., 2014). Sap flow was transformed to tree transpiration using individual tree sapwood basal area. Transpiration

at the stand level was obtained by multiplying the average tree sap flux density per unit sapwood area by mean tree sapwood cross-section area and the stand density. All data used for the evaluation are presented in Fig. 2. For further details regarding sap flow measurements see Klein et al. (2014).

Litterfall was collected in 25 litter traps of 0.5 m$^2$ each, along ten consecutive years (2003 - 2012). Litter was removed from the traps every 1–2 months and sorted into needle, reproductive, woody and residual fractions and oven-dried at 65 ˚C for two

days (Maseyk et al., 2008). For the purpose of this analysis, only needle litter was considered. To be able to compare simulated and observed dynamics, total leaf biomass was bootstrapped for the 2003 - 2012 period to derive the annual median (see Fig. S1) and then multiplied by an average needle longevity of 3 years (Maseyk et al., 2008).

SWC was monitored continuously at the site throughout 2013 – 2015. using Trime PICO-64 sensors (IMKO Micromodultechnik GmbH, Ettlingen, Germany) installed at depths of 5, 15, 30, 50, 70 and 100 cm in five soil pits, and

averaged over the whole profile. Air temperature and relative humidity were monitored continuously above the canopy at the flux tower (Tatarinov et al., 2016). The measured soil water content together with its representation by the model is given in the supplements (Fig. S2).

## 2.3 Model description

### 2.3.1 LandscapeDNDC and PSIM

LandscapeDNDC (https://ldndc.imk-ifu.kit.edu) is a simulation platform for terrestrial ecosystem models (Grote et al., 2011; Haas et al., 2013).It is designed to reproduce atmosphere-biosphere exchange process of carbon, water, and nitrogen, including trace gas exchanges. For this purpose, detailed soil process modules are provided to be coupled with ecosystem modules that are parameterized on the species level and cover grasslands, crops and forests. The LandscapeDNDC model framework uses daily maximum and minimum temperature, radiation, VPD and precipitation as meteorological inputs, which are downscaled

to hourly values. The canopy is divided into multiple layers which height and extension depend on the initialized ecosystem structure, and microclimate is calculated for each layer (using the Empirical Canopy Model, Grote et al., 2009)]. Similarly, soil is divided into a user-defined number of layers, each holding chemical and texture information (Holst et al., 2010). Foliage and fine roots are distributed across the canopy and rooting space, respectively, according to a distribution function that has been described in Grote and Pretzsch (2002).

The water balance is derived considering all major ecosystem fluxes (evaporation from interception, transpiration, ground surface and soil; runoff; percolation) and pools (water storage at the leaf surface, at the ground and in each soil layer) and is based on the original model for denitrification and decomposition (Li et al., 1992). The soil water content and distribution is basically represented with a bucket approach and soil water potentials are calculated based on soil properties using the equations suggested by Van Genuchten (1980). Forest carbon gain and losses by growth and maintenance respiration, as well

as phenology, allocation and senescence processes are considered within the Physiological Simulation Model (PSIM), which uses the Farquhar model to estimate hourly carbon assimilation (Farquhar et al., 1980), and is linked to a stomatal conductance module to optimize gas exchange. For standard simulations the procedure suggested by Leuning (1995) is applied, but alternative approaches are possible to select or to introduce (see below). Maintenance respiration is calculated based on temperature and nitrogen concentrations in the different tissues (Cannell and Thornley, 2000). The remaining photosynthates

are allocated into different tree compartments (reserves, foliage, fine roots, living wood) according to their respective sink strength, which is based on allometric relations (defines demand originating from foliage development), tissue loss rates (increases demand), and environmental limitations (preventing allocation to inactive tissues) (Grote, 1998). In case none of the compartments has any demands, the carbon is distributed according to allometric ratios between leaves, fine roots and sapwood (in case of undetermined growth) or between fine roots and sapwood (otherwise). Senescence of tree compartments

is generally derived from a specific longevity of each tissue. Currently an enhanced senescence of tissue under stressful environmental conditions is not considered.

In this configuration, LandscapeDNDC has been used to investigate gas exchange and biomass growth in forested ecosystems (Rahimi et al., 2021; Cade et al., 2021; Dirnböck et al., 2020). It has also been evaluated at different European forest sites (Mahnken et al., 2022; Nadal-Sala et al., 2021b) with one result being that the sensitivity of carbon and water fluxes to vapor

pressure deficit is generally not sufficiently well represented. Into this framework, we implemented a new hydraulic conductance scheme, as well a mechanism for stress-induced senescence of sapwood and foliage which is described in more detail below.

### 2.3.2 Representation of hydraulic conductance

- *Stomatal closure*

The newly implemented hydraulic approach into LandscapeDNDC allows the calculation of canopy water potential based on soil water potential and fine-root vertical distribution (see also Fig. 1a). Stomatal conductance ($gs$) is regulated in order to

optimize net photosynthesis at the one hand ($A_n$, µmol m$^{-2}$LA s$^{-1}$), which is calculated here according to Farquhar et al. (1980), considering a peaked Arrhenius response of photosynthetic parameters with leaf temperature (Medlyn et al., 2002), and hydraulic safety at the other, calculated from hourly mean canopy water potential following Eller et al. (2020):


$$gs = g_{MIN} + 0.5\frac{\partial A_{n}'}{\partial C_i}\left[\sqrt{\left(\frac{4\xi}{\partial A_{n}'/\partial C_i} + 1\right)} - 1\right] \tag{1a}$$

$$\xi = \frac{2}{\frac{\delta krc_{rel}}{krc_{rel}\ \delta\Psi_{\text{can\_mean}}}\ rp\ 1.6\ \text{VPDm}} \tag{1b}$$

$$rp = \frac{RPMIN}{krc_{rel}} \tag{1c}$$

$$krc_{rel} = e^{\left(-\frac{\Psi_{\text{can\_mean}}}{\Psi REF}\right)^{ACOEF}} \tag{1d}$$

$$\Psi_{\text{can\_mean}} = 0.5\left(\Psi_{\text{can\_PD}} + \Psi_{\text{canopy}}\right) \tag{1e}$$

Where $\delta A_n / \delta C_i$ is the increase in net photosynthesis per unit of internal carbon dioxide ($C_i$) increase – i.e. the gain function for stomata opening on net assimilation - while $\xi$ is the cost function, which represents the loss in hydraulic conductance with increasing stomatal opening. The function ensures that increases in $A_n$ are increasing stomatal conductance and vice versa,

while $gs$ is decreased with increasing plant resistance and vapor pressure (VPDm, mmol mol$^{-1}$). The single terms are the whole plant resistance to water flow ($rp$, m$^2$ s MPa mmol$^{-1}$), which is calculated from whole-plant minimum hydraulic resistance as defined in Eller et al. (2020) (**RPMIN**, m$^2$ s MPa mmol$^{-1}$), the relative root-to-canopy hydraulic conductance ($krc_{rel}$) (unitless), and the partial derivatives of $krc_{rel}$ and mean canopy water potential ($\Psi_{\text{can\_mean}}$). These are computed in hourly time steps as the linear gradient between $krc_{rel}$ ($\Psi_{\text{can\_mean}}$) and $krc_{rel}$ (0.5($\Psi_{\text{can\_mean}}$ + **$\Psi REF$**), respectively. $krc_{rel}$ in turn depends

on species-specific parameters (**ACOEF, $\Psi REF$**). $\Psi_{\text{can\_mean}}$ is assumed to be represented simply by the average of the predawn canopy water potential ($\Psi_{\text{can\_PD}}$) and the hourly calculated canopy water potential ($\Psi_{\text{canopy}}$, see Eq. 3) from the previous time step, to avoid abrupt drops in water potential along the plant hydraulic pathway (Eller et al., 2018). $\Psi_{\text{can\_PD}}$ is the value of $\Psi_{\text{canopy}}$ obtained directly before sunrise. Finally, stomatal conductance cannot decrease below a given minimum conductance ($g_{MIN}$), which represents canopy leakiness.

In order to enhance the impact of hydraulic constraints, we additionally consider a non-stomatal down-regulation of photosynthesis – hereafter referred as NSL - that has been suggested by various authors (e.g. De Kauwe et al., 2015a; Drake et al., 2017). We here assume a direct dependency to a declining $\Psi_{can\_PD,}$ using an equation suggested by Tuzet et al. (2003) and tested in Nadal-Sala et al. (2021a):

$$A_n' = A_n \left[ \frac{1 + e^{(\Psi NSL\ ANSL)}}{1 + e^{\left((\Psi NSL - \Psi_{\text{can\_PD}})ANSL\right)}} \right] \tag{2}$$

Where $\Psi NSL$ and $ANSL$ are species-specific parameters (see Table 1). The function results in decreases of the photosynthetic potential and thus to further reductions in stomatal conductance (according to equation 1a). The importance of this mechanism has been tested by running the model with and without the additional impact on photosynthesis.


- *Plant water potential and hydraulic conductance*

The relevant water potential for the canopy conductance control is $\Psi_{\text{canopy}}$, which is calculated from the xylem water potential following Darcy's law ($\Psi_{\text{xylem}}$, MPa) and canopy transpiration of the previous timestep ($T_{\text{canopy}}$, mmol m$^{-2}$LA s$^{-1}$) divided by root/canopy- or xylem hydraulic conductance ($Krc$, in mmol m$^{-2}$LA s$^{-1}$ MPa$^{-1}$), also considering the gravitational effect of

canopy height. The conductance term $Krc$ is obtained from the previous hour $\Psi_{\text{can\_mean}}$ and the species-specific xylem hydraulic vulnerability curve, assumed to follow a Weibull function (Neufeld et al., 1992):

$$\Psi_{\text{canopy}} = \Psi_{\text{xylem}} - \frac{T_{\text{canopy}}}{Kxylem} - h\rho g\,10^{-6} \tag{3a}$$

$$Kxylem = KSPEC\left[ e^{-\left(\frac{\Psi_{\text{can\_mean}}}{\Psi REF}\right)^{ACOEF}} \right] \tag{3b}$$


$KSPEC$, $\Psi REF$ and $ACOEF$ are empirically-determined coefficients (see Table 1) describing the shape of the vulnerability curve as has been obtained from field measurements (Wagner et al., 2022). The decline of $\Psi_{\text{canopy}}$ considers the gravitational impact with canopy height $h$ (m), where $\rho$ is water density at 25 °C (997 kg m$^{-3}$) and $g$ represents gravitational acceleration (9.8 m s$^{-2}$). The multiplication by $10^{-6}$ converts the term to MPa. In order to determine $\Psi_{\text{xylem}}$, the root water potential ($\Psi_{\text{root}}$,

MPa) has to be defined first from soil water potential ($\Psi_{\text{soil}}$, MPa) and vertical fine root distribution. $\Psi_{\text{soil}}$ is defined for each soil layer based on its water content, specific texture properties, and water holding capacity according Van Genuchten et al. (1991), with parameters determined according to Klein et al. (2014). Assuming that $\Psi_{\text{root}}$ equilibrates with $\Psi_{\text{soil}}$ overnight, it is generally calculated as the average $\Psi_{\text{soil}}$ of all ($n$) layers weighted by the respective fine root biomass fraction (De Kauwe et al., 2015b). The fine root distribution is described using an empirical function (Grote and Pretzsch, 2002) parametrized with

*in situ* data (Preisler et al., 2019). We also consider that under conditions of extremely low water potentials roots decouple

from the soil in order to prevent root-to-soil water flow (North and Nobel, 1991; Carminati et al., 2009; Carminati and Javaux, 2020):

$$\Psi_{\text{root}} = \sum_{i=1}^{n} frf_i \left( \max\left( \Psi_{\text{soil},i}, \Psi_{\text{disconnect}} \right) \right) \tag{4}$$


Where "$i$" indicates any given soil layer, and $frf_i$ is the relative root fraction per layer, and the species-specific water potential threshold at which the roots are decoupled from the soil is $\boldsymbol{\Psi_{\text{disconnect}}}$ (MPa). As long as $\Psi_{\text{soil}}$ is larger than $\Psi_{\text{disconnect}}$, transpiration demand is determining soil water uptake ($UPT$sw, mm), so that water demand and supply are assumed to be in equilibrium and $\Psi_{\text{xylem}}$ is equal to the water potential in the roots ($\Psi_{\text{root}}$, MPa). As long as $\Psi_{\text{soil}}$ allows, water uptake is distributed throughout
the soil layers according to fine root distribution and relative soil water availability. However, if the water reservoir within the core rooting zone is empty ($\Psi_{\text{soil}} \leq \Psi_{\text{disconnect}}$) remaining transpiration needs to be supplied by other sources such as the stem water storage. Accordingly, a tree water deficit ($WD$) develops cumulatively during the time without soil water uptake and is recovering as soon as $UPT$sw is again larger than $T_{\text{canopy}}$. Note that the implemented hydraulic processes do not principally limit $WD$. Therefore, a conceptual decision needs to be made by the model user to either consider trees to die once a critical
plant water potential has been crossed (e.g. at 88% percent loss of conductance (PLC), Liang et al., 2021), which would indicate that capacitance is depleted to a certain level (e.g. estimated to be about 30% of the living biomass dry weight, Ziemińska et al., 2020), or allow for water uptake from deep soil. In our case trees never reached such critical water potentials, likely due to a water supply from undefined deeper soil layers in accordance with earlier investigations at the site (Raz-Yaseef et al., 2010; Helman et al., 2017). This continued water uptake in the model however does not refill the depleted water sources in the trees,
but is solely supporting residual transpiration as long as ($\Psi_{\text{soil}} \leq \Psi_{\text{disconnect}}$). Hence, the trees continue to dehydrate and the water potential during this period ($\Psi_{\text{dehydration}}$, MPa) is calculated from the difference between $\Psi_{\text{can\_mean}}$ and $\Psi_{\text{root}}$ as follows:

$$\Psi_{\text{xylem}} = \Psi_{\text{root}} + (1 - fr)\, \Psi_{\text{dehydration}} \tag{5a}$$

$$\Psi_{\text{dehydration}} = \sum_{1}^{k} \left[ \left( \frac{\sum_{1}^{j} \left( \left( \Psi_{\text{can\_mean},j} + h\rho g 10^{-6} \right) - \Psi_{\text{root},j} \right)}{j} \right) \left( \frac{B_F}{B_S + B_R + B_F} \right) \right] \tag{5b}$$

$$fr = \max\left( 0, \frac{WD_{old} - WD}{WD_{old}} \right) \tag{5c}$$

$$WD = \max(0, WD_{old} + T\text{canopy} - UPT\text{sw}) \tag{5d}$$

Note that $\Psi_{\text{dehydration}}$ is an integrated term that increases throughout the period of "$k$" days as long as $\Psi_{\text{soil}} \leq \Psi_{\text{disconnect}}$. Since canopy water potential also includes a reduction by gravitational force (see Eq. 3) but $\Psi_{\text{dehydration}}$ is only expressing the dehydration effect, this term needs to be re-added to avoid double-accounting in the calculation of $\Psi_{\text{canopy}}$. The difference between corrected canopy water potential and $\Psi_{\text{root}}$ is then averaged over all daylight hours "$j$" per day. Additionally, a reduction term is required (Eq. 5b) that accounts for the fact that not all transpired water is drawn from the foliage but also from other living tree compartments (Tyree and Yang, 1990) . Therefore, we further assume that water capacitance is linearly related to biomass and that water deficits immediately equilibrate over all tissues (with $B_F$, $B_S$, $B_R$ representing foliage, sapwood and fine root biomass respectively, all in kg m$^{-2}$ ground).

- *Xylem inactivation and leaf senescence*

A new feature of the hydraulic module is the representation of the progressive loss of xylem functionality presented as sapwood area decline as $\Psi_{\text{xylem}}$ decreases (see Fig. 1; e.g. Choat et al., 2018; Rehschuh et al., 2020). The loss is calculated based on a hydraulic vulnerability curve represented with a Weibull function as:

$$\Delta_{\text{xylem,t}} = \min \left( e^{-\left(\frac{\Psi_{\text{can\_PD,t}}}{\Psi_{REF}}\right)^{ACOEF}} - e^{-\left(\frac{\Psi_{\text{can\_PD,t-1}}}{\Psi_{REF}}\right)^{ACOEF}} , 0 \right) BA_{\text{xylem}} \quad (6)$$

Where $BA_{\text{xylem}}$ is the basal area of xylem per tree (m$^{-2}$ tree$^{-1}$), $\Delta_{\text{xylem,t}}$ is the daily reduction in xylem basal area (m$^{-2}$ tree$^{-1}$), and $\Psi_{\text{can\_PD,t}}$ and $\Psi_{\text{can\_PD,t-1}}$ are the canopy pre-dawn $\Psi$ at the present day and the previous day, respectively (both in MPa). **ACOEF** and **$\Psi$REF** are species-specific parameters (see Table 1). This process is not directly reversible, with xylem functionality being regained only via regrowth of new tissue (see Fig 1b) (Hammond et al., 2019). Finally, in this model version, leaf area is reduced proportionally to $\Delta_{\text{xylem,t}}$ / $BA_{\text{xylem}}$, according to the pipe model (Shinozaki and Yoda, 1964).

**2.3.3 Model initialization and parameterization**

Parameters for the LandscapeDNDC core processes such as the species-specific temperature sum that determines leaf flushing, the electron transport rate under standard conditions that defines photosynthesis, allometric relations and tissue longevities that drive allocation and senescence were obtained from the literature (Bernacchi et al., 2001; Infante et al., 1999; Kattge and Knorr, 2007; Medlyn et al., 2002; Navas et al., 2003) (see supplementary Table S2 for *P. halepensis* parameters). Note that some of these parameters are derived directly at the investigation site including the ones describing photosynthesis (Maseyk et al., 2008), foliage biomass and development (Zinsser, 2017) and fine root distributions (Klein et al., 2014). Therefore, absorption properties of canopy and rooting space, which dimensions are defined by the stand and soil inventory information

(see 'Site description'), are actually based on measurements. Also, some parameters that were used for the new hydraulic scheme, such as the ones related to xylem vulnerability (Wagner et al., 2022), were available from observations at the Yatir forest (Table 1).

With the exception of the loss of conductance parameters (*ACOEF, ΨREF*) and the residual conductance term $g_{MIN}$, which both were taken from literature, the hydraulic parameters for the new module were calibrated using an inverse Bayesian calibration (Hartig et al., 2014; Dormann et al., 2018) based on GPP measurements for the 2013 - 2014 period (Table 1). We implemented a Gaussian likelihood function, with a 'Differential Evolution with snooker update' algorithm as a sampler (DEzs, ter Braak and Vrugt, 2008). 50k simulations were run for the calibration, with a burn-in of 30k simulations. The three chains of the calibration had converged at this point, i.e. Gelman-Rubin score for all marginal posteriors < 1.1 (Gelman et al., 2013). The LandscapeDNDC simulations were then run with the median values for each calibrated parameter. Adjusting parameters to GPP only, has the advantage that we could use sap flow and water potential measurements for evaluation. On the other hand, the procedure bears considerable uncertainty about the system behaviour throughout the whole range of environmental conditions observed. In order to ensure that all parameters can be varied within reasonable boundaries without getting unrealistic impacts, in the following we will investigate the sensitivity of transpiration and plant water potential to $g_{MIN}$ and all calibrated parameters with standard values used for all other parameters (see supplementary Figs. S3, S4). Priors and credible intervals for each parameter were selected within literature boundaries broad enough to allow the model to capture the responses, but sufficiently constricted to bound them to biologically meaningful limits. For example, the median of $g_{MIN}$ for woody species was reported as 3.0 mmol m$^{-2}$ leaf area (LA) s$^{-1}$ for *P. halepensis* but the range for semi-arid plants is given by Machado et al. (2021) as 1.1 - 6.3 mmol m$^{-2}$ LA s$^{-1}$ (see Table S3 for the prior distribution).

### 2.3.4 Statistical analysis

All analyses were performed using the software R, version 4.1.2 (R Core Team, 2021). The parameter calibration of LandscapeDNDC was done using the "BayesianTools" package (Hartig et al., 2019). When a Type I linear relationship has been applied, simulated vs. observed evaluation was given as Spearman's $R^2$ and Root Mean Square Error (RMSE). To assess the relationship between measured sap flow – as a proxy for transpiration – and modelled daily plant Ψ gradient (ΔΨ$_{plant}$, in MPa), calculated as Ψ$_{can\_PD}$ - min(Ψ$_{can,mean}$) a Type II linear regression was implemented using the "segmented" package (Muggeo and Muggeo, 2017). To derive the threshold at which the modelled daily Ψ$_{canopy}$ became uncoupled from VPD but driven by SWC we performed a truncated linear analysis with the "BayesianTools" package.

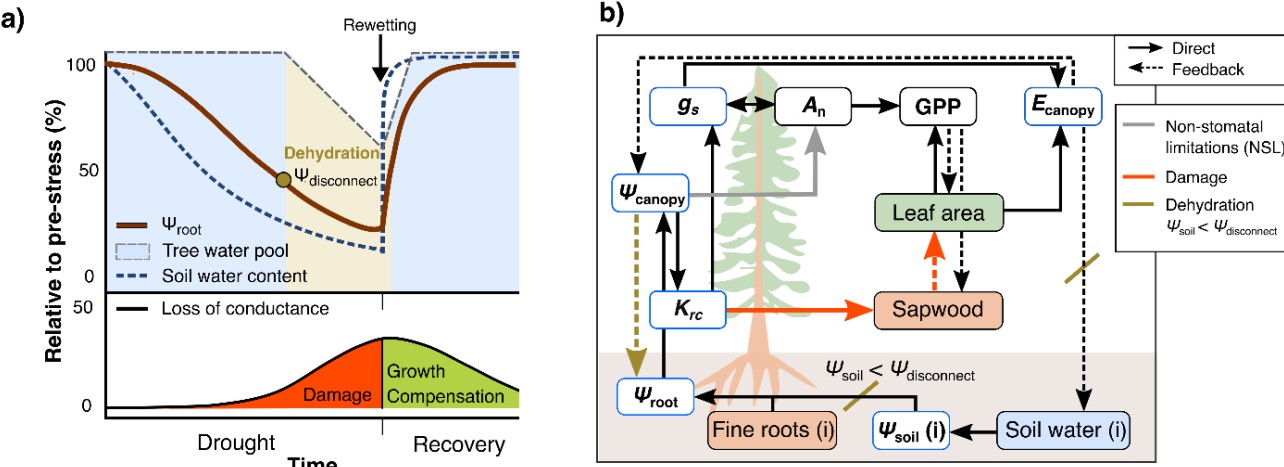

**360**

**Figure 1: Conceptual scheme of the hydraulic approach implemented into the model framework.** (a) Theoretical progression of drought and recovery alongside SWC dynamics and its relative impacts on root water potential and tree water pool. Once the root water potential ($\Psi_{root}$) falls below a threshold, roots disconnect from the soil ($\Psi_{disconnect}$) and trees begin to dehydrate, emptying an internal tree water pool. During this stage, the functional damage to the trees caused losses of hydraulic conductance. Following re-wetting, functional impairment
**365** is slowly reversed via regrowth of foliage and sapwood area. (b) Schematic overview of hydraulic processes, including the decrease of photosynthetic capacity ($A_n$) (non-stomatal limitation, NSL) and thus gross primary productivity (GPP), in turn affecting stomatal conductance ($g_s$) and transpiration ($E_{canopy}$). The root-soil disconnect ($\Psi_{soil} < \Psi_{disconnect}$) is highlighted, triggering tree dehydration and biomass loss induced by declining root-to-canopy hydraulic conductance ($Krc$).

**370**

**Table 1: Key parameters for the new hydraulic module in LandscapeDNDC for Aleppo pine applied to the Yatir forest.** Parameters have been derived from Bayesian calibration and the literature. For the Bayesian approach, median and the 95% credible intervals (CI) per parameter are given.

| Equation | Parameter | Unit | Median [CI] | Description | Source |
|---|---|---|---|---|---|
| 1 | $RPMIN$ | mmol$^{-1}$ m$^2$LA s MPa | 3.8 [2.8, 4.4] | minimum whole-plant resistance | Bayesian |
| 1 | $g_{MIN}$ | mmol H$_2$O m$^2$LA s | 3.0 | minimum leaf conductance | (De Cáceres et al., 2023) |
| 2 | $ANSL$ | Unitless | 3.5 [3.2, 3.8] | curve parameter for effect on $A_n$ | Bayesian |
| 2 | $\Psi NSL$ | MPa | -1.01 [-1.06, -1] | reference $\Psi_{canopy}$ for effect on $A_n$ | Bayesian |
| 3 | $KSPEC$ | mmol m$^{-2}$LA s$^{-1}$ MPa$^{-1}$ | 1.9 [1.7, 2.5] | Specific xylem conductance | Bayesian |
| 4 | $\Psi_{disconnect}$ | MPa | -1.75 [-1.56, -1.95] | $\Psi_{soil}$ threshold of soil- root disconnect | Bayesian |
| 1, 3, 6 | $ACOEF$ | Unitless | 7.5 | curve parameter for $\Psi_{canopy}$ impact on conductance | Wagner *et al.* (2022) |
| 1, 3, 6 | $\Psi REF$ | MPa | -3.8 | reference $\Psi_{canopy}$ for conductance vulnerability curve | Wagner *et al.* (2022) |

## 3 Results

### 3.1 Model evaluation

The simulated GPP dynamics after model calibration captured the observed GPP pattern with a Pearson's correlation coefficient around $R^2 \sim 0.8$ (Fig. 2). The impact of the summer drought on tree water relations was reflected in the GPP dynamics with lowest uptake rates during the driest period (April-October). This clearly represents a huge improvement over previous versions of the LandscapeDNDC model (see Fig. S5) and indicates the suitability of the newly implemented hydraulic processes to capture GPP dynamics particularly during extreme drought. The agreement was only slightly higher during the period used for the Bayesian parameter calibration than when strictly comparing to the evaluation period, indicating a low bias in the calibrated parameters. Despite the good overall fit, particularly covering the steep decline after the rainy season, GPP seems to be underestimated in the dry period, particularly during the first year. However, since the deviation is considerably stronger in the first year, temporally restricted impacts deriving from the model initialization, such as available surplus water from previous year, actual leaf biomass, or spatial redistribution of water originating from rainfall events not covered in the data set are likely influences (Shachnovich et al., 2008).

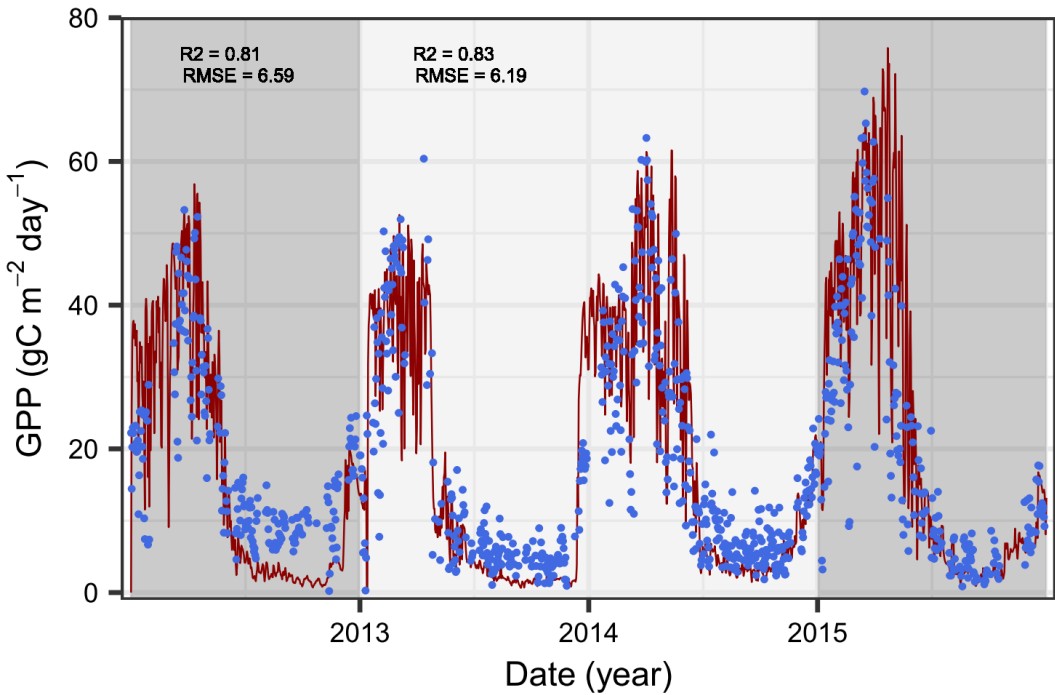

**Figure 2: Comparison of simulated (red lines) and observed (blue dots) gross primary production (GPP) in Yatir forest.** The periods used for the Bayesian calibration (2013-2014, light grey) and model evaluation (2012 and 2015, dark grey) are highlighted and the goodness of fit between model and observations are indicated as the Pearson's correlation (R2) and the root mean square error (RMSE).

### 3.2 Sensitivity of tree water relations to seasonal drought and VPD

The model simulations captured the strong seasonality of water availability at the Yatir forest with mild and relatively wet winter conditions and dry summer periods. This was reflected in the modelled predawn plant water potential ($\Psi_{can\_PD}$) ranging between -0.7 MPa during the wet winter season and -2.3 MPa during the dry summer period (Fig. 3a, black line). For evaluation, we compared occasional plant water potential measurements during the years 2012-2014 with $\Psi_{can\_mean}$, which varies during the day (Fig. 3a, grey area). Except of one event at the onset of the dry period 2013, which showed particularly low values, simulations covered all measured potentials within the uncertainty ranges. It should be noted that the daily variability of $\Psi_{plant}$ decreases considerably when approaching a plant water potential of -1.75 MPa (= $\Psi_{disconnect}$), after which no additional water is taken up and the daily cycle is only driven by redistribution of water within the plant. During this time, further tree dehydration depends strongly on $g_{MIN}$, VPD and leaf area (see also Fig. S3 and S6). The modelled daily gradient in water potential ($\Delta\Psi_{plant} = \Psi_{can\_PD} - \min(\Psi_{can\_mean})$) over the three simulation years was in high agreement with the observed transpiration rates (Fig. 3b). Further we found good agreement of simulated and observed transpiration rates (Fig. 3c), while properly reproducing soil water content (SWC) dynamics (see Fig. S2).

The simulations indicate that internal tree hydraulic dynamics are overall limited by soil water availability and that the decrease is steeper after a threshold at about 15.8% SWC (95% CI [15.4, 16.5]) has been reached, which is well before $\Psi_{disconnect}$ (Fig. 4a). As long as the availability of soil water is above the threshold, transpiration is more sensitive to changes in VPD while
below this threshold the sensitivity to SWC is more strongly expressed (Fig. 4b). Hence, during the wet season, stomatal conductance depends mostly on evapotranspiration demand, while in the period of soil drying conductance is mostly limited by soil water availability.

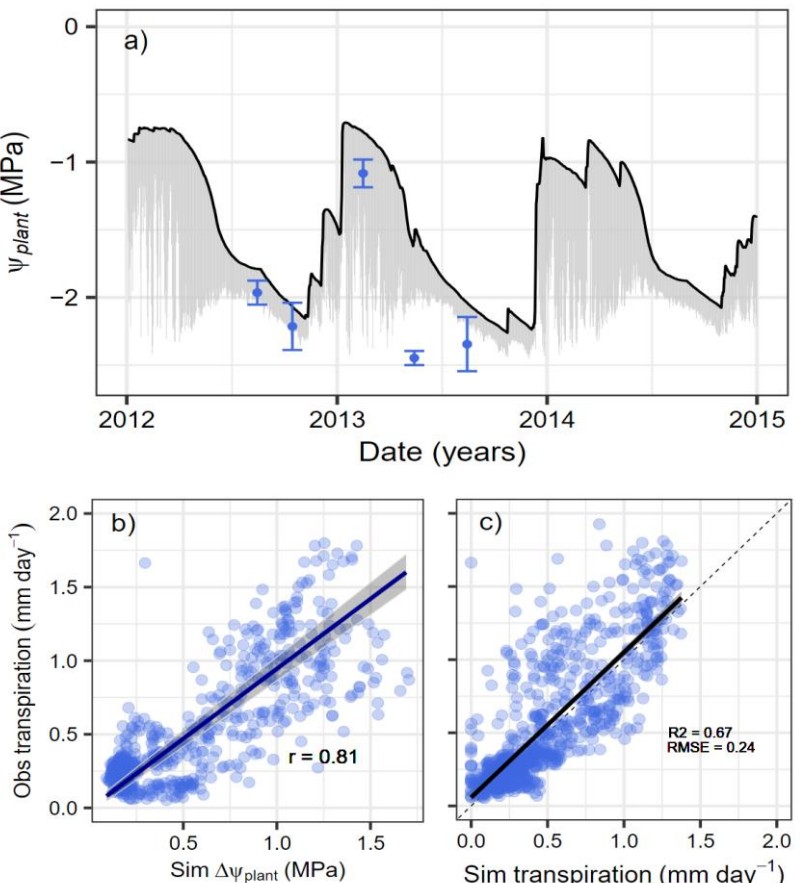

**Figure 3: Simulated and observed water potential and transpiration responses to seasonal drought at the Yatir forest.** Dynamics of simulated plant water potentials ($\Psi$can_PD, $\Psi$can_mean, $\Psi$measured) and transpiration rates with observations at Yatir forest from 2012 to 2014. a) Seasonal dynamics in pre-dawn plant water potential (black line) and the daily water potential gradient (grey area). Observations represent midday leaf water potentials (blue circles) with uncertainty ranges given ($\pm$ SD) as reported by Preisler et al. (2019). b) Relationship between simulated $\Delta\Psi_{plant}$ (see text for explanation) and observed transpiration rates are given with Pearson's correlation coefficient for a
Type II linear regression. c) Comparison of observed against simulated daily transpiration rates.

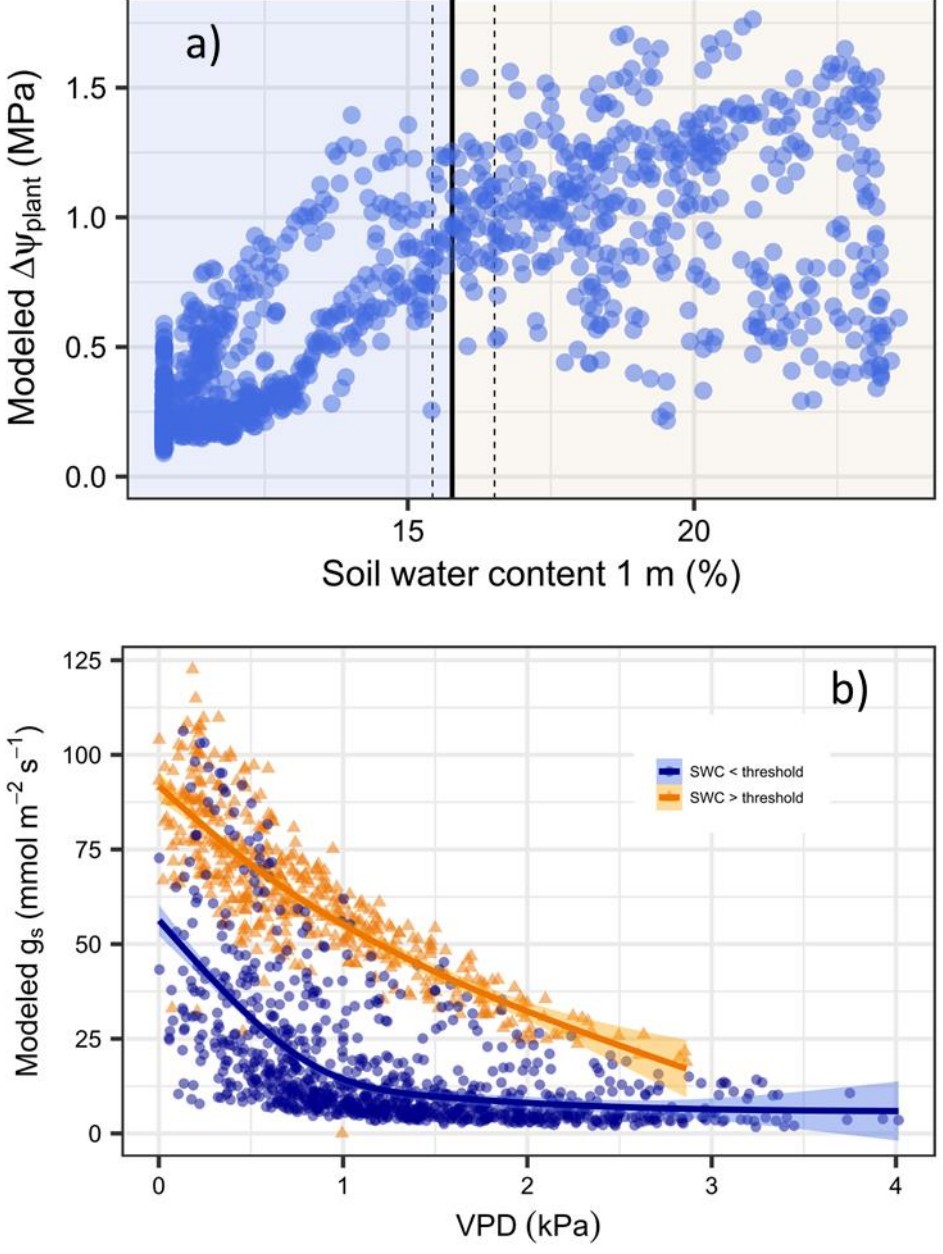

**Figure 4: Identification of dominant drivers for plant pre-dawn water potential (Δ$\Psi_{plant}$) in Aleppo pines at the Yatir forests.** a) Δ$\Psi_{plant}$ in relation to water content within the rooted soil (SWC, in %). Vertical lines indicate the SWC at which a shift in the driver dominance occurs (solid line = median, dashed lines = 95% confidence intervals). b) Relationships between daytime daily averaged stomatal conductance and daytime daily averaged VPD for SWC above (orange triangles) and below (blue dots) the SWC threshold.

### 3.3 Sensitivity of tree water relations to non-stomatal limitations

The sensitivity of simulated water fluxes to specific processes has been investigated by testing responses of transpiration and plant water potential to variations of hydraulic parameters (Fig. S3). Since the sensitivity to the process of non-stomatal limitation depends on various parameters and model assumptions, we tested the impact of early, moderate, and late onset of the NSL impact (Fig. S4). The respective simulations demonstrate that an early onset of photosynthesis decline decreases $gs$ and transpiration considerably faster than a late onset and is able to prevent plant water potentials from reaching damaging levels (Fig 5). Without considering the direct limitation on photosynthesis, $gs$ responses to $\Psi_{\text{can\_PD}}$ are delayed and stomatal closure was consequently reached at an unrealistic low water potential for an isohydric species such as Aleppo pine.

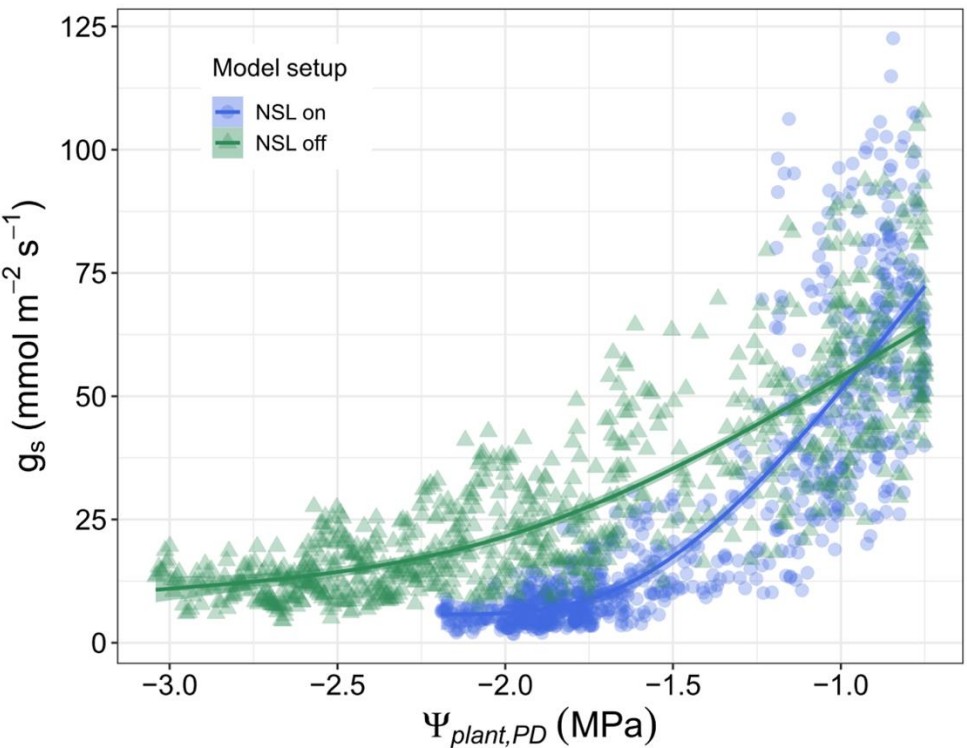

**Figure 5: The impact on non-stomatal limitations (NSL) on $gs$.** Stomatal conductance derived from model runs with and without the NSL routine versus pre-dawn plant water potentials ($\Psi_{\text{can\_PD}}$) during the dry-down from March to August for all years (2012 – 2015).

## 3.4 Hydraulic impairment and leaf shedding

Based on the evaluated plant water potentials, the additional stress-induced loss of xylem area according to Eq. 6 is accumulating to 3.4–6.3% per year of total sapwood area. The net loss of sapwood, which is composed on tissue loss by age as well as by low plant water potentials - occurs solely during the dry season when allocation to sapwood is zero or close to zero (Fig. 6, brown areas). It starts at $\Psi_{can\_PD}$ values of approximately -1.25 MPa, far away from both P12 (-2.9 MPa) and P50 (-3.6 MPa) - i.e. the plant water potential at which a 12% and a 50% of xylem conductance has been lost, respectively (see Fig. S4). The additional stress-induced loss of conductance, albeit relatively small, is responsible for the differences in functional sapwood area during the dry seasons of the different years. During the wet season, sapwood growth generally copes with - or even exceeds- - the demands for foliage supply (determined by the sapwood area / foliage area ratio, Table S2) and is thus positive (Fig. 6, green areas).

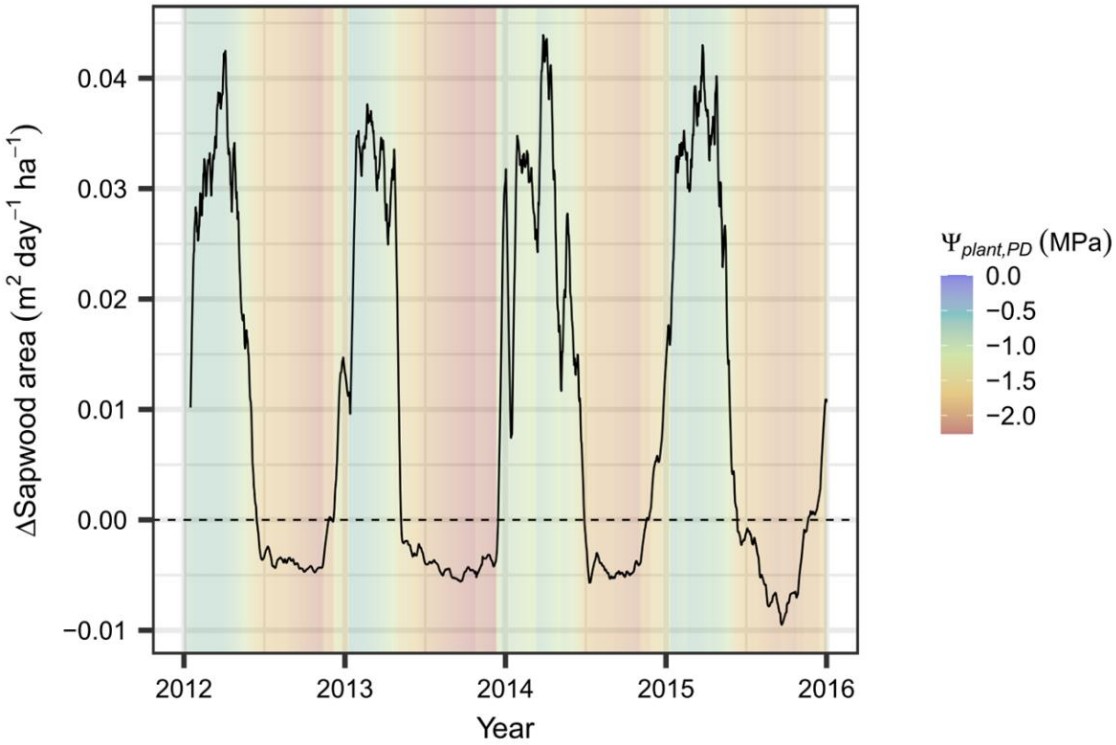

**Figure 6: Simulated sapwood area dynamics at Yatir forest.** Simulated net gains and losses of active sapwood area (ΔSapwood area) are presented as 7-day moving average during 2012 - 2015. The corresponding daily predawn plant water potential is given as coloured background area.

The overall pattern of foliage litterfall reproduced the observed seasonal dynamics reasonably well (Fig. 7). In our model, flushing and phenological leaf senescence are determined to start by the onset of the wet period in January and end by mid-September, closely after the onset of the dry period. According to our model concept, functional xylem losses translate into

460 additional foliage losses during the dry season, resulting in stress-induced litterfall between September and December. However, similar to the relatively small amount of drought-induced loss in sapwood area it was only small.

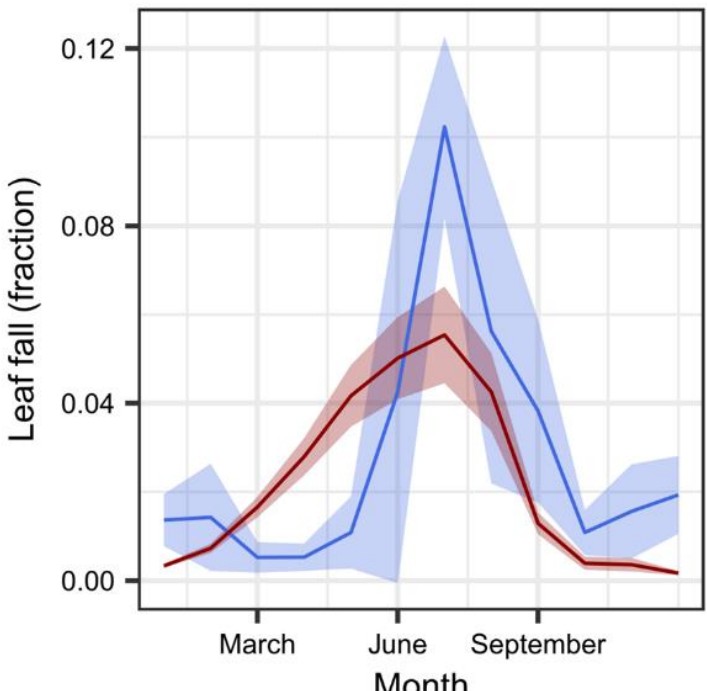

**Figure 7: Seasonality of leaf litterfall from observations in the Yatir forest and respective simulation results.** Simulated (red) and
465 observed (blue) monthly median litterfall is shown as a fraction of average leaf biomass. Note that simulations are from 2012 - 2015, while observations were integrated from 2003 - 2012. The shaded areas represent the 95% CI.

### 3.5 Sensitivities of drought-induced tissue senescence to $g_{MIN}$ and root-soil disconnection

Hydraulic damage in the model is mostly restricted to the period after the roots become disconnected from the soil, and
470 dehydration during this period depends largely on residual evaporation ($g_{MIN}$). Therefore, we have tested the sensitivity of the model to variations in the two parameters $g_{MIN}$, and $\Psi_{disconnect}$. The selected range was determined based on published values for conductance under dry conditions (Klein et al., 2011; Llusia et al., 2016) and observed ranges of predawn water potentials

in in *P. halepensis* at different sites (Atzmon et al., 2004), in order to highlight its pivotal role in the new model formulation. Over the observed range, an increase of both, $g_{MIN}$ as well as $\Psi_{disconnect}$ results in a linearly increased sapwood area damage

and percent loss of conductivity (Fig. 8). The selected parameter combination does fall in the lower range of conductivity loss that has been observed at the site (Fig. 8, grey area), at the same time, the analysis shows that the tissue damage is sensitive in particular to $\Psi_{disconnect}$ and can easily be under- or overestimated.

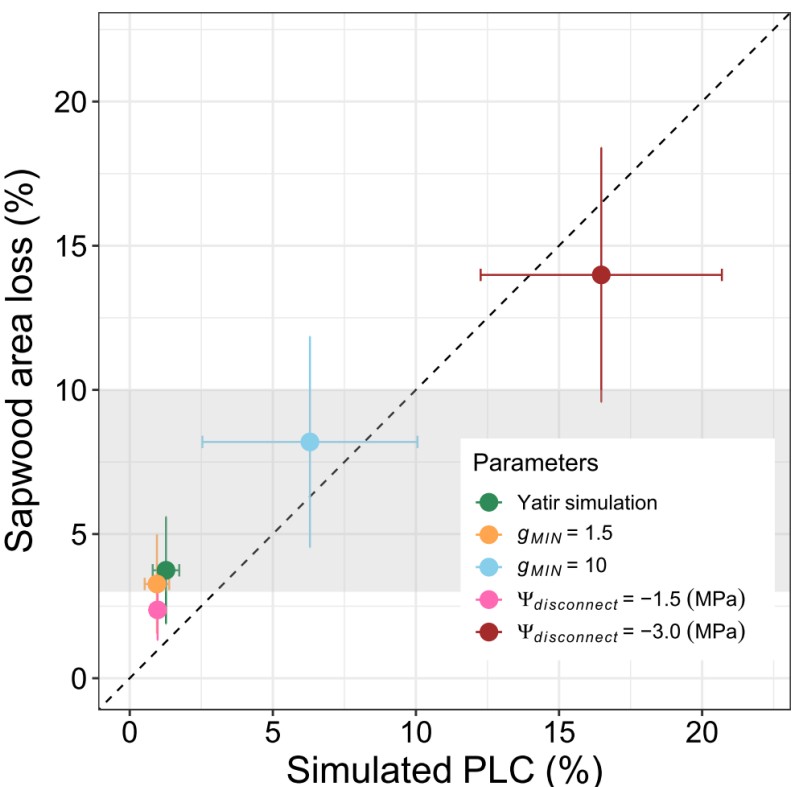

**Figure 8: Impacts of changes in hydraulic key parameters $g_{MIN}$ and $\Psi_{disconnect}$ on sapwood area loss.** Simulated sapwood area loss is shown as annual averages obtained during the 2012 - 2015 period in relation to simulated maximum percent loss of conductance during summer (PLC, in %). For comparison, sapwood area losses of about 3-10 % as reported in Wagner et al. (2022) and Feng et al. (2023) for the 2020 - 2021 period, are indicated as grey shaded area.

## 4 Discussion

Our simulations indicate a tight coordination of drought-induced physiological responses in a seasonal summer dry forest that are triggered by a decreasing plant water potential affecting stomatal closure, soil root-disconnection and tissue senescence. In the selected case study, *P. halepensis* shows an expressed isohydric behaviour which is in close agreement with previous observations (Fotelli et al., 2019; Klein et al., 2011). Accordingly, a seasonal differentiation in the importance of environmental drivers on stomatal conductance was clearly apparent with fully open stomata during the rainy season and gradually declining conductance down to a very small minimum towards the end of the dry period (see also Fig. S7). We could show that VPD was the main influence to stomatal behaviour during the period with sufficient soil water supply, whereas during the rest of the year SWC was limiting gas exchange. When soil water content was close to its minimum, $g_{MIN}$ is the most important parameter for the dehydration processes and hence tissue damage under prolonged drought stress. Our simulations, which are in close agreement with sap flow measurements, indicate that transpiration could not be supported solely from soil water within the assumed rooting zone during the peak of the dry season. The water additionally required was in the range of approximated capacitance (few mm) during 3 of the 4 investigated years, but increased cumulatively up to 25 mm in one of the investigation years. We suppose that the supply from deeper soil layers is the most likely explanation for this which is in line with earlier investigations (Raz-Yaseef et al., 2010). Even assuming that the damage to xylem and needles is linearly related to the PLC curve development, the model indicates only minor damage to the Aleppo pines in Yatir, which demonstrates the effectiveness of physiological counter measures such as the early onset of NSL impact that slows down the decrease of the internal water potential, or the high resistance to xylem damages as indicated by the PLC curve.

The simulations are capturing all water fluxes within a reasonable range. Water losses by transpiration are of about two thirds of precipitation, interception and soil evaporation cover about 11% each, and the remaining water is lost to percolation during high rainfall events (see Table S4). This means that model estimates are similar with respect to interception and transpiration but smaller regarding total evaporation if compared with eddy-flux measurements documented by earlier studies at this site (Raz-Yaseef et al., 2010; Ungar et al., 2013). One reason may be that soil evaporation has been underestimated by the model maybe originating from a faster water transport away from the soil surface than what is actually happening at the site. Also, water adsorption from the air might play a role under semi-arid conditions with temperatures below condensation during night (Qubaja et al., 2020). The latter leads to water condensing at plant surfaces during the night that is evaporating in the morning and thus adds to evaporation but is not considered in the model. Overall, percolation is simulated to be considerably higher (36 mm in average over the 4 years simulated) than the additional water needed to supply transpiration when upper soil water is depleted (maximum of app. 20 mm in 2013), supporting the model assumption that the trees are able to take up water from deeper soil layers, a mechanism that has already been assumed a this site (Preisler et al., 2019).

## 4.1 Drought stress mitigation due to enhanced stomatal sensitivity

Within the hydraulic scheme, the stomatal conductance mechanism needs to account not only for various drivers but has to consider each of them appropriately during any specific phase of stress. For example, stomata regulation under humid to moderately dry condition is most sensitive to VPD (Novick et al., 2016; Tatarinov et al., 2016). This sensitivity is captured well with the introduced hydraulic approach which decreases $gs$ under increasing VPD to avoid excessively low water potentials (question i). The enhanced sensitivity of stomata is particularly triggered by the consideration of photosynthesis down-regulation under drought stress. This is a mechanism well suitable to represent isohydric behaviour, which is supported by the finding that NSL effects are indeed particularly observed in isohydric species such as birches, poplar and pines (Uddling et al., 2005; Salmon et al., 2020). Under the selected extremely dry conditions the sensitivity to NSL is combined with a rather insensitive (or resistant) conductance loss curve, which ensures the survival of the species. However, it is not yet clear if this relationship between NSL and vulnerability could be scaled with the drought sensitivity in general.

The observed shift in Aleppo pines' water management from demand (VPD)-driven to supply (SWC)-limited driving factors is best represented by accounting for a very sensitive direct impact of drought stress on assimilation. Such a consideration has been demonstrated to be particular suitable under conditions of a steep decline in water availability (question ii). The integration of the NSL effect considerably enhances stomatal sensitivity to drought via a feedback mechanism from limited photosynthetic carbon uptake (Flexas and Medrano, 2002; Tissue et al., 2005; Gallé et al., 2007; Zhou et al., 2013). Hence it enables a very steep response and avoids the necessity of a rather unrealistically low water potential under drought stress (Sabot et al 2022). Hoshika et al. (2022) found an important role of photosynthesis downregulation for deciduous as well as evergreen oaks, and Wilson et al. (2000) estimated that this mechanism is responsible for approximately 75 % of the stomatal regulation in several deciduous trees. Nevertheless, the sensitivity of this mechanism is certainly species-specific (Lobo-do-Vale et al., 2023) and may strongly vary with foliage age (see e.g.Wilson et al., 2000). Consequently, accounting for this NSL effect has been considered recently in various models (Dewar et al., 2022; Nadal-Sala et al., 2021a; Salmon et al., 2020) and might be essential to represent fast stomatal, particularly isohydric responses to high evaporation demand (e.g. Yang et al., 2019; Gourlez de la Motte et al., 2020).

The vulnerability curves previously established for *P. halepensis* all indicate a medium sensitivity for a declining plant water potential with 50% of the conductance lost at about -3.5 to -5 MPa (Gattmann et al., 2023; Oliveras et al., 2003; Wagner et al., 2022). However, soil water potentials in Yatir can easily drop below -10 MPa in the upper layers (Klein et al., 2014). To avoid plant dehydration but still using the measured vulnerability curves, we therefore defined that the roots disconnect from the soil (calibrated at about -1.75 MPa), which lead to a slow-down of the water potential decrease and conductance loss, and also a slow-down of soil water depletion. The threshold obtained by Bayesian calibration is very close to the approximately -2 MPa reported by Klein et al. (2014), also indicated as wilting point of the upper soil. This strengthens our notion that changes in root-to-soil resistance are very important to constrain the hydraulic damage of a drought stressed plant (question iii). The importance of root detachment from the soil when calculating the increase of resistance with decreasing soil water has been

found to account for more than 95% of total plant hydraulic resistance (Rodriguez-Dominguez and Brodribb, 2020) supporting our assumptions. The process is increasingly recognized and implemented into hydraulic models (e.g. Lei et al., 2023). This is based on the experimental evidence that roots loose contact from the soil under dry conditions (Carminati et al., 2009; Rodriguez-Dominguez and Brodribb, 2020), which has been suggested as a determining factor for soil water depletion, slowing down soil drying (Carminati and Javaux, 2020). In fact, considering soil-to-root decoupling using a $\Psi_{soil}$ threshold can be seen as a simplification of more complex models, which simulate a steep root-to-soil conductance decline explicitly (Cochard et al., 2021; Sperry et al., 2017; De Cáceres et al., 2023). In the current approach, actual water compartments within the tree are only very coarsely considered to achieve plant water potentials less negative than soil water potentials. We have used a further simplification by assuming a non-specific water supply from internal tree water storages or deeper soil which is necessary to sustain transpiration during summer without a further depletion of soil water reserves, which agrees with *in situ* observations (Preisler et al., 2019). Complex plant hydraulic models are avoiding these simplifications by calculating more water pools explicitly and using a higher tissue-level resolution, thus providing a high degree of realism, but introducing more uncertainties related to parameterization of soil layer composition, tree compartment conductivity and hydraulic segmentation, fine root distribution, and fine root size and density (e.g. Cochard et al., 2021; Haverd et al., 2016).

Overall, the simulations clearly demonstrate the importance of **residual water loss** via leaf leakiness and bark transpiration in tree dehydration processes (Márquez et al., 2021; Machado et al., 2021). This is in agreement with previous work highlighting the importance of stem internal water reserves for the survivorship of Aleppo pine in the Yatir forest during the summer dry season (Preisler et al., 2022).

## 4.2 Consideration of structural hydraulic constraints

Although we were not able to refer simulated loss of sapwood area directly to measurements, independent observations at the same site indicate that the projected sapwood area reduction of up to 6% is close to *in situ* branch embolism observed at Aleppo pine trees during summer, which were indicated as 8–10% (Wagner et al., 2022). Other reported values for *P. halepensis* embolism that have been measured near Tel Aviv, Israel, indicate less than 5% of functional loss at the end of summer (Feng et al., 2023). Albeit Aleppo pines showed to be well-adapted to the extreme drought conditions, an increase in critical stress damage can still be expected under future hotter and drier conditions that increase the residual water loss and may amplify hydraulic damage, particularly under high VPD conditions (e.g. Wagner et al., 2022). Similarly, leaf area loss was within observed limits of litterfall, although could not be evaluated directly with the available data. This is partly due to the small difference between litterfall due to needle longevity and stress-induced senescence, and partly originates from the time lag between the death of needles and actual litterfall. To better evaluate the approach, model applications at other long-term observations sites in dry regions [e.g. in France or Italy (Reichstein et al., 2002)] will be required.

Based on the result at the quite extreme site in Israel, we think that the model opens the possibility to address prime legacy impacts of drought stress through linking tree hydraulics to stress-induced leaf senescence and sapwood inactivation (question

iv). Stress-induced structural adjustments have been identified as a stress response signal, resulting in a reduction of drought vulnerability, for instance by decreasing leaf area available for evaporation through hydraulic segmentation (Hochberg et al., 2017; Wolfe et al., 2016). On the other hand, tissue losses are costly and if not replaced by reserves as soon as conditions are favourable again, will lead to a lower 'income' of carbon. Thus, the introduction of a mechanistic representation of sapwood and foliage mortality provides the possibility to integrate hydraulic failure and carbon starvation into a unified model framework. That is, on the one hand, the loss of hydraulic conductivity provide means to trigger tree death directly based on a threshold of xylem damage beyond which regrowth is impossible (Hammond et al., 2019; Trugman et al., 2018). On the other hand, an increasing demand of carbon to regenerate conductive woody tissue will lead to a shortfall of supply for building new leaves (and roots), reducing tree C uptake and potentially inducing a long-term decline process under recurrent drought stress. If a tree will be able to survive and recover or if the additional carbon demand will result in its delayed death, depends on the balance between resource supply and demand, which are both strongly influenced by stand structure, competition and climatic boundary conditions (Camarero, 2021).

### 4.3 Further model developments

In order to better capture forest responses to increases in extreme events, process-based models such as LandscapeDNDC that are integrating micro-environmental, physiological and tree growth processes are important tools - not only to project carbon and water fluxes but also to implement mitigation efforts of forest management. Here we consider two main avenues of further model developments, which include a better description of tree capacitance as well as an explicit characterization of tissue senescence responses to drought. Both are not only limited by modelling capabilities but also by limitations of our current understanding.

Our modified model allows for residual transpiration after full stomatal closure that originates from undefined water sources (as described above), such as **tree capacitance or deep soil-water access**. While the importance of such water reserves is generally undisputed (e.g. Gleason et al., 2014; Ripullone et al., 2020) and has recently been found as a major determinant for survival or decline (Schmied et al., 2023), the supply of water from the plant tissue is nevertheless limited and thus needs to be constrained. The limitation on water storage depends on structural variables, namely stem dimension (Zweifel et al., 2020) and wood traits (Christoffersen et al., 2016), but its availability may also be described dynamically, e.g. in dependence on xylem activity [e.g. water is released only after cavitation occurred, Hölttä et al. (2009)]. In turn, the dehydration rate is determined by leaf leakiness, incomplete stomatal closure and bark transpiration, among others (Duursma et al., 2019). Hence, the residual conductance $g_{MIN}$ and the stem water capacitance are key for tree survivorship as drought progresses (Blackman et al. 2019). Since $g_{MIN}$ is not directly affected by the increase in water potential, a logical next step is to link the depletion of stem water content to the process of hydraulic failure (Scholz et al., 2011). Alternatively, residual conductance and capacitance impacts may be implemented dynamically, for example in dependence on air temperature (Schuster et al., 2016), which has

been recently realized in the SurEau model available within the MEDFATE model package (De Cáceres et al., 2023). Besides this, to our best understanding none of the processes mentioned above are considered in ecosystem- or forest models yet.

**Drought-induced defoliation** has been addressed with the suggested model approach but it is questionable if foliage dynamics should respond to sapwood dynamics following the pipe model theory (Yoda et al., 1963). The reasons are twofold: First, it is unlikely that an irreversible xylem damage occurs already at low water potentials where the representation of PLC already indicates damages, albeit very small ones. It would thus be logical to introduce some species-specific thresholds accounting for xylem structure and stability (Gauthey et al., 2022). Second, the current event chain does not allow for preventive leaf shedding. An alternative has been proposed with the "hydraulic fuse" hypothesis Hochberg et al. (2017), which postulates a direct dependence of leaf shedding to plant water potential in order to capture protective acclimation processes to drought (Wolfe et al., 2016; Li et al., 2020), with hydraulic segmentation among different tree compartments at its core. However, the benefit of losing leaves depends on the costs of rebuilding new ones, and also on the xylem hydraulic failure risk at faster declining water potentials. Thus the tendency to shed leaves protectively may be less expressed in trees with higher leaf longevity (Mediavilla et al., 2022), or in the exceptional case of trees being able to refill the embolized vessels (Choat et al., 2018). Also, we still lack empirical evidence for mechanistic description general enough to link leaf senescence to xylem water potential decline. Deriving such relationships in a range of tree species would provide exciting possibilities for further model development. Aleppo pines -albeit considered as relative drought-resistant species – have been shown to die by hydraulic failure once a specific threshold of xylem disfunction has been reached (Morcillo et al., 2022).

The implementation of a hydraulic strategy into a process-based model allows for feedback responses and enables to represent individual responses of different species, or evaluate the suitability of different leaf shedding strategies under future environmental conditions. Such an approach could be used to investigate for example the benefit of a high resistance strategy with costly tissues (e.g. evergreen species, high wood density), compared to a highly vulnerable species with tissues less costly to reproduce (Saunders and Drew, 2022).

## 4 Conclusions

In order to capture drought impacts on forest functioning, models need to address both, the quickly reversible physiological responses as well as the slowly reversible impacts from drought-induced functional impairment and structural damage. This is highly challenging since they need to be addressed in good agreement with the available observations while adhering to the maximum parsimony principia. Example simulations based on the presented hydraulic approach could demonstrate that early non-stomatal impacts are able to effectively reduce the desiccation process and delay the onset of more severe damages. Also, an increased resistance at the root/soil interface seems to be supporting the principle of reducing the water uptake capacities to prevent more severe damages. Following this principle, we propose that the suggested simple representation of plant water potential is also able to drive long-term responses leading to structural adjustment and thus represent legacy impacts of drought

on plant function. We consider it a prerequisite for a mechanistical representation of a progressive decline of tree functioning and consequent mortality following recurrent and intense droughts. The mechanistic link between stress-induced damages from hydraulic impairment and carbon allocation processes may be suitable to shed new light on the hydraulic failure and carbon starvation continuum resulting in tree mortality.

**Supplement link** < to be provided >

**Software and model code**

The LandscapeDNDC model source code for released versions of the model is permanently available online at the Radar4KIT database (https://doi.org/10.35097/438; Butterbach-Bahl et al., 2021). The published model version has been used as the basis for integration of the new hydraulic scheme is "win64 ldndc-1.30.4" which can also be freely downloaded upon request from the following website: https://ldndc.imk-ifu.kit.edu/download/download-model.php (last access: 25. August 2023). The new model options described in this paper are also documented in the online-model description ([https://ldndc.imk-ifu.kit.edu/doc/ldndc-doxy.php](https://ldndc.imk-ifu.kit.edu/doc/ldndc-doxy.php), assessed 25.08.2023). All input data to run the model are either freely available from the internet (see site description) or are provided in the supplement (soil properties, initial stand properties, species-specific parameters).

**Author contribution**

DN-S, RG and NKR designed the conceptual approach, determined the modelling setup and led the manuscript writing. RG, DN-S and DK coded the hydraulic module into LandscapeDNDC. DN-S performed the data analysis. DY, FT, UH, YW and TK contributed to the field measurements at Yatir forest and provided the observational data. All co-authors contributed to writing and revising the manuscript.

**Competing interests**

The authors declare that they have no conflict of interest

**Acknowledgements**

This study was supported in parts by the German Research Foundation through its Emmy Noether Program (RU 1657/ 2-1), its German-Israeli project cooperation program (SCHM 2736/2-1 and YA 274/1-1) and the German Israeli Foundation (GIF

grant 1539). NKR acknowledges funding through the Helmholtz Initiative and Networking fund (W2/W3-156). We also acknowledge support by the KIT-Publication Fund of the Karlsruhe Institute of Technology. Finally, we are thankful to Yakir Preisler, Eyal Rotenberg, and Josef Gruenzweig for field support.

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
