# Peer review of "Integration of tree hydraulic processes and functional impairment to capture the drought resilience of a semi-arid pine forest"

_Biogeosciences, 2023_

## Author Comment (AC2)

Supplement to response to Reviewer 2 for submission:

Integration of tree hydraulic processes and functional impairment to capture the drought resilience of a semi-arid pine forest

[Figure]

Figure R1: Determining safety margins with different NSL response curves. Assuming that soil-root disconnection occurs only after NSL response has reached a certain threshold (here 95%, approximately correlating with stomata closure), different NSL responses (defined by parameters A_NSL and PSI_NSL) are representing different safety margins until loss of tissue conductance (PLC curve).

[Figure]

Figure R2: Sensitivity of predawn plant water potential to different thresholds of soil-root disconnection. Using the thresholds defined previously from NSL sensitivity analysis, different stress-related states and fluxes (here: predawn-water potential) are obtained for the Yatir forest site during the investigation period (2012 to 2015).

---

## Author Response (AR1)

General comments regarding the new manuscript version

We are thankful for the insightful review and the general appreciation as an approach that can be considered interesting and important for further research. We have used the comments for improving the manuscript along the lines indicated below. In particular, we tried to avoid the impression that the approach is too empirical to be useful in general. In fact, the implementation into LDNDC has been done with the objective to be developed into a general application to which many tree species or plant types can be parameterized for.

The three main concerns expressed by the reviewers were that i) too many parameters were calibrated, ii) the evaluation is relatively weak, and iii) the impact of the new implementation is not very well illustrated. Also, it was particularly requested to better connect the article to the current literature and also to consider available hydraulic approaches presented in the past.

i) Regarding the first issue, we first have to apologize for the sometimes-confusing presentation of parameters which we have now improved in the current manuscript. Overall, only five parameters were actually calibrated, 2 of them describing the shape and onset of the direct photosynthesis impact (NSL effect), and 3 are representing different plant resistances, including $\Psi_{disconnect}$ which can be seen as an indefinite resistance that only applies under particularly dry conditions. We think that in the future we can derive these parameters from gas exchange measurements and soil water potential estimates also for other species. In order to justify our approach, we would like to emphasize that alternative modelling concepts often use much more detailed processes and hence more difficult to define parameters, smaller time steps and higher spatial resolution (e.g. conductivity for different plant parts and heights), which then are considerably more difficult to parameterize and to apply in general.

ii) We admit that the evaluation of the presented approach is concentrating on few continuous as well as discontinuous measurements. It is thus strongly limited by the available data, in particular the Eddy-covariance flux measurements of carbon exchange to calibrate the model, and the independent sap flow, soil water content, and occasional water potential data to evaluate the behavior of the model regarding transpiration. Nevertheless, applying the concept at one site only, with simulations covering not more than a few years, lets space for an extended model description while the evaluation is still strong enough to provide a proof of concept, which is not unusual in similar publications. Still, the evaluation could have been better explained and discussed which we try to do more comprehensively in the current manuscript version.

iii) At many places in the new text we now better illustrate the benefits and impacts of the processes newly implemented into LandscapeDNDC. Particularly, the dependency of an isohydric, risk avoiding strategy on the direct limitation of photosynthesis through non-stomatal limitations. Therefore, we have added more sensitivity analyses that cover this NSL effect as well as the parameters $g_{MIN}$ and $\Psi_{disconnect}$ which also demonstrate the improvement in fit obtained with the new model. In addition to the already available Figure 8 and S5 in the manuscripts and the supplements, respectively, this includes the addition of Figures S3 and S4.

In addition, we are now acknowledging previous work on hydraulic model development more comprehensively and tone down anything that may have given the impression of complete novelty. The actual improvement is rather the consideration of various hydraulic impacts that are considered in a relatively simple but consistent manner than providing a new process itself.

The point-by-point response below intends to describe the details of all mentioned manuscript improvements.

Detailed responses (A) to reviewer 1 (R1)

R1: The manuscript added plant hydraulics' influence on ecosystem carbon and water fluxes into the LandscapeDNDC model. The new model was calibrated and evaluated for a pine plantation in Israel. The manuscript presented the multiple pathways of plant water stress (stomatal, non-stomatal, leaf shedding, and sapwood loss) and their importance in simulating drought impacts. Overall, I like the idea of organizing plant water stress as a sequential physiological response triggered by plant water potential, as summarized by many eco-physiological studies cited in the paper.

However, I feel the modeling approach presented is **too empirical to make the added processes generalizable** and useful (maybe constrained by the model structure of LandscapeDNDC). Particularly, the **only real evaluation** of the plant hydrodynamic module is pre-dawn water potential, which is highly determined by soil hydraulics instead of plant hydrodynamics. In addition, there are various **modeling efforts (mostly in the context of tropical forests) that have implemented sequential responses, which are not acknowledged**. Altogether, the essentiality of the added module is not well highlighted and these reduce the significance and novelty of the study.

A: We are grateful for expressing the impression of a model approach not very well evaluated and difficult to generalize. In fact, our purpose was to find a relatively simple yet physiologically sound solution for the problem of representing hydraulic impairment on various levels. As a result, we suggest an approach that is following general mechanistic rules for stomata closure, photosynthetic damage, and morphological responses. This approach is embedded in a system that allows the application to any tree species with only few parameters to be additionally determined. Besides a more precise language we also add Figure S4 with a conceptional scheme demonstrating the relation between photosynthesis downregulation, vulnerability of conductance, and plant water potentials.

It is true that the hydraulic module evaluation was done using relatively few pre-dawn water potentials only, but the overall model was also evaluated with independent measurements on transpiration and soil water content. In the current manuscript we emphasize this issue and highlight the overall good fits with additional descriptions of sensitivity to parameter variations (apart from Figure 5 and 8 see also Fig. S3 and S4).

In addition, we have taken care to acknowledge previous modeling approaches of hydraulic constraints, which now forms a new paragraph in the introduction. In fact, all processes presented have been used in other models before albeit hardly in combination.

R1: The introduction claims that plant hydraulic processes are not represented in a consistent way in ecosystem modeling (Line 50-55, Line 85-90, etc.). This is not true. Christofferson et al. 2016 and Xu et al. 2016 have both fully integrated plant hydrodynamics with plant physiology in demography-explicit ecosystem modeling. In particular, Xu et al. 2016 implemented stomatal, non-stomatal (through a reduction in carboxylation capacity), and phenological responses to drought. Meanwhile, explicit tracking of sapwood dynamics is indeed rare. Most of the modeling practices implicitly include the reduction of sapwood fraction through a reduction in conductivity.

A: We apologize for giving the impression that plant hydraulics have not been introduced in models before. Of course, this is not true. We are now giving an overview of such approaches (Introduction, L55ff) stating the suggested references as well as a number of others (Kennedy et al., 2019). We also introduce work that deals with the simulation of non-stomatal processes (Gourlez de la Motte et al., 2020; Keenan et al., 2010; Yang et al., 2019) and morphological impairment (Yao et al., 2022). Still, we argue that models struggle between simplicity and parsimony and are in many cases not yet considering stomatal, non-stomatal, and morphological impacts together (which are the ones that can lead to better capture drought legacy responses in forest growth). Overall, it is more the simplicity and the connection between physiology (stomata) and structural dependencies (sapwood dynamics), and the potential they offer to represent lagged tree responses to drought, rather than the innovative hydraulic representation we aim to, which is now also expressed in our newly formulated objectives (L129ff).

R1. Many of the hypotheses (Line 108-114) are not appropriate to be answered by a modeling study. For instance, testing the first hypothesis on VPD limitation on stomatal conductance would require physiological observations. If one codes in VPD effect in the model, you will surely see VPD effects independent from soil moisture. Furthermore, isn't the VPD effect already known as early as the Leuning or Collatz stomatal model (if not earlier) ... Similarly, the second and third hypotheses need some more thought. One way to postulate useful hypotheses in a modeling study is to test whether a new process shifts model behavior (and under what circumferences) and make model results closer to observations. Hypothesis iv reads better and more interesting compared with other ones.

A: We are grateful to the reviewer for pointing this out. We have therefore reformulated the objectives considering the suggestions (L129ff). For the first objective we concentrate on the general evaluation while pointing out that a particular challenge is the consideration of both the VPD as well as the soil drought effect in balance. It is supposed that this is an integrated task not self-fulfilled by the implementation of one process. Also, the second and third objectives were reframed, now targeting specific processes (non-stomatal impact, and root disconnection from the soil).

R1: I have a hard time understanding and interpreting **psi_dehydration** (line 250-255). First, shouldn't psi_xylem be equal to psi_root minus sapflow/transpiration divided by root-to-xylem resistance? I am not certain about the motivation for defining psi_dehydration. Second, psi_dehydration is the average potential gradient between canopy and roots, weighted by foliar biomass, in the past k days. Which is then used to calculate instantaneous canopy water potential later in equation (6a). The mismatch of time-scale (k-day average vs instantaneous) is really confusing and I am confused about the downregulation of the gradient by foliar biomass. I have some other concerns on the hydraulic physiology of the model specified later.

A: We admit that the description of the dehydration term was difficult and also not completely correct. The improved description can be now found in Equation 5a-d (now also including the connection to the water deficit) and the surrounding paragraphs (L283ff, L301ff).

The transpiration term is not considered in psi_xylem but in the calculation of psi_canopy, which is now moved upwards in the description as Equation 3. Here, it is also clear that the water potential is indeed calculated by a difference in potential divided by a resistance term and

that the gravitational pull is considered. Consequently, we needed to compensate for this pull-term in Equation 5 in order to avoid double accounting (now written in order to foster intuitive understanding).

The confusion with psi_dehydration likely originates from the fact that it is only considered during severe drought (i.e. when the disconnection threshold has been passed). The integration over k days is caused by the increase of dehydration at every day where the water deficit is increasing due to residual transpiration. The term is accompanied by a damping function which is based on the logic that water for transpiration is supplied first from the plant tissue (capacitance effect). The more water in the plant, the more the decrease of water potential in the plant is dampened.

R1: I have two main concerns over the results and discussions. First, there are a lack of evaluation/benchmark of the new plant hydraulic module. The only evaluation is Fig. 3a, which shows the model can predict seasonality and diurnal cycle of water potential. **However, the model seems to significantly overestimate water potential in the dry season of 2013**. I understand hydraulic measurements might be rare to get. Some more discussions on modeled hydrodynamics would be helpful.

For example, what is the **diurnal cycle** of transpiration/sapflow (I think these are the other hydrodynamic variables available at the site?) compared with the observed sapflow in the wet season and dry season respectively.

A: We apologize for an obviously unclear description but the seemingly overestimation of water potentials in Fig. 3a probably results from a misunderstanding. It should be noted that the black line indicates the simulated predawn potentials while the measurements are done during the day, mostly at midday. They are thus represented by the grey shaded area. We have improved the respective description in the mostly rewritten paragraph above (L393ff). Admittedly, there is still one mismatch in May 2013 where the model seems to behave too conservative compared to the plant water potentials. An explanation is difficult, particularly because these measurements are indicating more drought in May than in July, which seems counter-intuitive and is not backed up by any rainfall in between. We are therefore cautious to blame the model for the mismatch.

Although the data on water potentials that are used for evaluation of the simplified drought mechanism are not particularly comprehensive, we consider also the sap flow measurements as evaluation data, which are more or less continuously available over each day of the investigation period, and they provide an independent dataset to evaluate the model outputs against. These are corroborating an overall good match between simulations and measurements (Fig. 3c).

Following the suggestion above, we have added a figure demonstrating the diurnal cycle and its development during soil water depletion into the supplementary (Fig. S7). This nicely complements the improved discussion of the overall water balance that has been added in the new manuscript version (L484ff, L500ff).

R1: Second, there is a lack of comparison for models with/without the new additions (except for Fig. 5 when comparing NSL on and off). **It is not clear to me how the new additions are essential to correctly model transpiration, GPP**, etc. For example, if calibrating the original LandscapeDNDC with MCMC, could the calibrated model capture seasonality in transpiration and GPP?

A: Thank you for your suggestions. Although varying single parameters is strictly speaking not appropriate in a Bayesian approach (because other parameters would be differently estimated each time), we admit that the sensitivity of processes and fluxes to specific parameters can best indicated this way. Therefore, we accompanied the simulations with or without consideration of NSL by simulations where we varied the most sensitive parameters GMIN and PSI_DISCONNECT and presented the impacts on transpiration as well as plant water potential (Figure S3). This is nicely illustrating and supporting the impact such a variation on sapwood area loss (Figure 8).

In addition, we also illustrate the impact of different NSL strategies, expressed as an early, moderate or late photosynthesis decline in response to plant water potential. This nicely shows that the investigated pines are responding before any significant damage on tree conductance occurs and that this strategy is protective regarding further water pressure declines. In turn, a later response is prolonging high transpiration rates and decreasing plant water potentials down to damaging levels (Figure S4).

Besides the 4-year GPP development (Figure 2) - which is the result of literature parameterization and hydraulic parameter calibration - and the evaluation of transpiration (Figure 3c), the supplementary sensitivity figures (S3 and S4) also function in addition to Figure 5 to demonstrate the improvements achieved with the new hydraulic module.

Technical Comments:

R1: Line 58, 'Already' is excessive here

A: deleted (L78)

R1: Line, 75-80, As I mentioned above, there have been many models that simulate plant hydraulics at stand-scale, including but not limited to ED2 (Xu et al. 2016), ELM-FATES (Christofferson et al. 2016), CLM (Kennedy et al. 2019), JULES (Eller et al. 2018), and ORCHIDEE (Yao et al.2022). There are challenges while much progress has been made already.

A: We have added a new paragraph to the introduction with a more comprehensive overview about hydraulic approaches already considered in models using the references mentioned above and complementing them with few others (L62ff). In particular the history of non-stomatal limitations and their representations is noted. We are taking care not to provoke the impression that we are presenting a new approach but rather working with simplified assumption covering several impacts consistently.

R1: Line 120-125 What would be the soil water potential for the **wilting point**? This might be related to the 'disconnect' water potential.

A: Indeed, the wilting point should be principally similar to the soil water potential of disconnection. If the van Genuchten Parameters are well set, soil water potentials at wilting point should be so low that water uptake gets impossible – which is what our introduced disconnection threshold is expressing. Therefore, the current model implementation is disregarding the wilting point in the site file as a limit for water uptake. This relationship has been described in the text (L146ff) and a note has been added in Table S1.

R1: Line 143-145, given there is an EC tower as well. I wonder whether the sapflow-based transpiration has been compared with tower-based ET?

A: It is true that an eddy tower is installed at the site. We have compared part of these data with the model output which indicated that the model underestimated total evapotranspiration. Likely reasons are an underestimation of soil evaporation because the model might lose soil water from the upper layers too quickly into deeper ones, and also lacks to consider adsorbed water during the night that evaporates during the day (Qubaja et al. 2020). However, due to the high evaporative demand at this site, the impact on transpiration and thus physiological responses is only small. We added this aspect to the discussion of the total simulated water balance which has been newly added to the manuscript (L484ff).

R1: Line 169, is the soil carbon/nitrogen module relevant here? If not, it can be removed.

A: The sentence has been removed (L196).

R1: Line 205-210, I am confused by the variable RPMIN and the calculation of rp. What do they represent physically? In addition, both RPMIN and krc_rel have a unit of mmol/m2/s/MPa, so how could their quotient also have a unit of mmol/m2/s/MPa?

A: We are grateful to the reviewer for pointing out the mismatch in units. In fact, krc_rel is unitless (a relative unit) so that Equation 1c is matching. RPMIN is the whole-plant minimum hydraulic resistance, which is an integrated resistance value that we use as defined by Eller et al. 2020. We have tried to elaborate the description in the text accordingly (L236ff).

Line. 215-220. So, wdef is some kind of magical residual soil water pool that plants can access in the dry season? This means the hydrological budget of the model is not conserved. How negative wdef can get?

A: Indeed, wdef (or as it is now termed WD) can be seen as an undefined residual water pool, which is necessary to introduce once a minimum conductance larger than zero is considered (and also necessary to supply the measured transpiration rates when constraint by upper soil water content). In our simulations this pool can accumulate up to app. 20mm per year from which stem water storage might supply a few mm only (which is, however, sufficient in some years). More than an undefined water pool WD is a measure of stress (in form of a water deficit experienced by trees) that provides a metric to which tree mortality can be linked. Therefore, a threshold water potential or PLC will trigger tree death and thus prevent indefinite additional water supply in future model versions. The derivation and dynamics are described with new paragraphs (L284ff in methods and L484ff in discussion) and the additional equations 5c and d.

R1: Line 279 'hydraulic vulnerability cure' --> hydraulic vulnerability curve

A: misspelling corrected (L313)

R1: Line 319, Fig. 1, **Soil evaporation** is missing in the figure. Is it important in the ecosystem?

A: This is true. The reason is that the figure doesn't show the water balance but the plant hydraulic approach and its direct influences. This doesn't include soil evaporation as well as evaporation from interception or percolation into deeper soil layers. Including all water balance terms would be possible but has been considered in order not to overload the figure. Soil evaporation, however, is now given in the supplementary Table S4.

R1: Line 345, Fig. 2, the simulated GPP is biased low in 2012-2013 dry season. What could be the potential explanations?

A: Indeed, despite GPP is the target for calibration, the fit between simulated and measured GPP deviated towards the dry period. Since the deviation is considerably stronger in the first year, we assume that temporally restricted impacts deriving from the model initialization or spatial redistribution of water originating from rainfall events not covered in the data set are likely influences (in a bit more elaborated way, this has also been put into the text, L381ff).

R1: Line 385, Fig. 5, It is great to show model behavior difference. A further question is which one is closer to observations. Is there any way to benchmark these two curves with observations? Maybe you can plot and contrast GPP vs soil moisture for the two simulations as well as observations?

A: Thank you for this suggestion. We also had the feeling that the mechanism of the direct photosynthesis impact needs a bit more elaboration. Therefore, we introduced the supplementary figure S4, where we illustrate the impact on different NSL sensitivities (see Fig. S4a that is also given below in response to reviewer 2) as well as the impact of including or excluding such process to plant water potential and transpiration. In addition to Figure 3c, where it is shown that the whole model is meeting the transpiration very well, we can show here that transpiration (and thus GPP) is better represented using the NSL method than discarding it. The effect is particular important for representing the fast decline on photosynthetic activity during late-spring, towards dry periods (see also elaborated description in section 3.3., L427ff).

R1: Line 401, Fig. 6, very interesting figure and I really like the implementation of sapwood turnover and growth. Just curious does the sapwood area increment match the observed tree basal area growth at the site?

A: Unfortunately, there are no tree growth records for this period. We also looked at other inventories but all suffered on the fact that different in tree samples were taken. Based on comparisons with single similar sized trees, LandscapeDNDC seems to slightly overestimate the increase in woody biomass and thus dimensional growth. For longer-term estimates it will be important to check also the allocation rules of the model in order to estimate if full recovery can be achieved also after very stressful conditions.

R1: Line. 470-480, Prieto et al. 2012 has discussed about asymmetric root-soil hydraulic conductivity, which might give rise to the disconnect water potential.

A: We have improved the discussion about the realism of the simplified disconnection process based on literature (L540ff). We are thankful for the suggested reference but have decided against it favoring e.g. the more specific and more up-to-date reference (Rodriguez-Dominguez and Brodribb, 2020)

Detailed responses (A) to reviewer 2 (R2)

R2: This manuscript presents and evaluates new developments made on the LandscapeDNDC modelling framework, focused on improving the realism and model performance under drought stress, using data from an extremely dry Aleppo pine plantation. The design of model modifications is sound and addresses several issues that are at the forefront of current modeling efforts in the community. The resulting model seems to perform appropriately, at least according to the observed data sets available. Furthermore, the discussion of the importance of the different processes and representation is interesting. I like Fig. 1 and how the manuscript contribution is framed, in general.

My only main concert, however, is on the way objectives are stated, which in my opinion is a bit odd. In particular, I don't think the questions targeted are those stated. For example, question (i) has an obvious answer, yes, as it is only a matter of model design. Actually, there are other models that separate the influence of soil water potential and leaf water potential on transpiration/photosynthesis. Is the authors' objective to gain knowledge of the importance of the different processes in the ecophysiology of plants? or to be able to successfully represent those processes in a model framework? and evaluate the sensitivity of model predictions to their representation? I believe the authors seek the second objective, but the current text seems to navigate between both and is not clear in this respect.

A: We understand the concern of the reviewer that the objectives are not targeted enough, which agrees with the concerns risen by reviewer 1. Indeed, we are searching to represent the relation between plant water potential and conductance with consistent and relatively simple to handle mechanisms. Therefore, we are happy to change the objectives accordingly, also covering the aim for evaluation and testing the model at an example site.

Following this, our revised objectives are: i) to evaluate the newly developed plant hydraulics module at an extreme seasonal dry forest site. In particular, the module is challenged to represent the two main seasonal trends in Yatir regarding stomatal behaviour: VPD-driven stomatal limitation during times of ample soil moisture and soil moisture-driven limitations under dry environmental conditions. ii) to determine the potential importance of hydraulic-driven non-stomatal limitations on photosynthetic assimilation; and iii) to assess the impact of considering a root-to-soil disconnection process under realistic conditions of prolonged drought stress. Furthermore, we depict and discuss how the proposed hydraulic modelling scheme could be used to alter simulated leaf and sapwood area dynamics. (L129ff)

R2: In my opinion, by adding more clarity in the end of the Introduction, and in the Discussion section, the overall manuscript would improve in usefulness. I have a suggestion related to this. Besides evaluating the model performance with observed data, I suggest the authors to more

straightforwardly compare the effect of including the different modifications (hydraulic model, NSL, defoliation) one by one, as done for NSL. With these comparisons, the reader will understand the importance of considering these processes, in the LandscapeDNDC model framework or others.

A: We appreciate the suggestion of adding analyses more targeted to the specific processes. It is not an easy task to address each process independently because the stomatal control model and the non-stomatal influences as well as the root-soil disconnection interact on a short time scale with each other. Thus, each time one process is cut out, parameters that are adjusted with a calibration process only would need re-calibration. 1). Nevertheless, we have introduced new sensitivity analysis for the threshold of root-soil disconnection, and GMIN, and have elaborated on the description of the NSL effect. Therefore, Figures S3 and S4 were introduced into the supplementary, demonstrating the sensitivity of plant water potential as well as transpiration to different parameter settings or to switching off the process completely. This is complemented with an illustrative explanation of how an early, moderate or late onset of photosynthesis shutdown will either save the plant of low water potentials (in case of early responses) or develop into the range where PLC losses are expected (see also S4a below).

[Figure]

Figure S4a: Illustration of impacts of different NSL response curves. Assuming that soil-root disconnection occurs only after NSL decline has reached a certain threshold (here 95%, approximately correlating with maximum stomata closure), different NSL responses (defined by parameters PSI_NSL) are representing different safety margins until loss of tissue conductance (PLC curve).

Along with these illustrations, we improved the description, in particular in section 3.2. (L395ff) and 3.3. (L427ff) in order to better describe and thus clarify the impact of single

processes: stomatal conductance, non-stomatal impact on photosynthesis, and soil-root disconnection.

Defoliation and sapwood loss, however, is inherently connected to the empirically determined loss curve on conduction. It is currently assumed that tissue loss happens along this curve without any thresholds, meaning that it is not using specific process-related parameters (see improved discussion L610ff). Available data at this site are not sufficient for evaluation and therefore, representing tissue mortality needs to be seen as a potential further development of our module which we illustrate yields reasonable results.

*Minor comments*

R2: L47 – Non-stomatal limitations to photosynthesis.

A: 'transpiration' changed into 'photosynthesis' (L54)

R2: L154 – What about precipitation data? Did came from a gauge in the same tower?

A: Precipitation data have been taken at the flux tower, indicated in L154

R2: L207 – Please add more details on the formulation of the cost function (xi: $\xi$).

A: The description of the cost function has been improved with corrections of the units, and clarifying sentences that describe its relation to vapor pressure deficit and previous water potentials (L236ff) and also includes further references to literature.

R2: L209 – is krc_rel a relative root-to-canopy hydraulic conductance or an absolute one with explicit units? Not clear.

A: Indeed, it is a relative unit and has been changed accordingly (L231). Thanks for pointing this out.

R2: L215 – Not clear how gmin relates to wdef. Do you mean that gmin increases wdef progressively, once stomata are closed?

A: What is meant is that the speed of water deficit accumulation (now WD) is larger with a larger GMIN (and vice versa). The irritating sentence has been deleted and the development of WD is explicitly described adding equation 5d (L299).

R2: L228 – Are further reductions in stomatal conductance due to An' affecting eq. 1? If so, mention this for clarification.

A: We are now mentioning the feedback to equation 1 explicitly (L255).

R2: Eq. 4 – I would add a 'Delta' symbol to Psi_dehydration, since it seems a water potential drop, rather than a water potential value.

A: This is a possibility. However, the dehydration is a cumulative value and only a fraction of it is put in use to decrease the xylem water potential. In order to clarify this, Equations 4 and 5 have been merged and complemented with the impact of the water deficit (new Eq. 5a-d)

R2: L255 (eq. 5) – I was expecting this equation to relate Psi_dehydration to wdef explicitly, but it does not. Then, wdef accumulates because of gmin and eq. 6a? The way internal water redistribution affects dehydration is not clear either. I would expect PV curves (relating water content to water potential) be used here.

A: As mentioned above, Equations 4 and 5 have been merged and the impact of the water deficit is explicitly related to it as suggested (new Eq. 5a-d). The redistribution effect has not been changed but the explanation for the simplified solution has been elaborated (L301ff).

R2: L271 (eq. 6b) – Can these parameters be estimated from standard vulnerability curves?

A: Yes. In fact, ACOEF and PSI_REF both are determined from measured and published PLC curves. We expect that many other species can be parameterized this way from existing literature. KSPEC is slightly different since this conductance term might actually vary throughout the tree system. Our simplification thus demands a 'representative' value that may be a weighted average of different conductance along the pipe. Still, it might be derived with empirical measurements.

L304 – Why is Vcmax,25 not mentioned in Table 1? It was calibrated but is not considered a key parameter?

A: We apologize for the mistake. Vcmax25 has been set by literature since direct measurement data were available (correctly put into table S2) but it was NOT calibrated (thus now deleted in table S3). This was based on a previous model development phase where the work of (Kuusk et al., 2018) was not known.

L311 – "BayesianTools"

A: Spelling corrected (L352, L356)

L360-363 – There are some inconsistencies in this interpretation. If the turning point is behaviour is psi_disconnect, how is it possible that after disconnection stomatal conductance is mostly limited by soil water availability. Then, you state that the dehydration rate depends on gmin and VPD, whereas the evapotranspiration demand was mostly affecting conductance during the wet season. Please, revise these sentences.

A: Thanks for pointing out some sources of irritation. We shortened the respective paragraph here in the result section and elaborated on the interpretation in the discussion section, which already included the issues (section 4.1).

L442 – You could be more specific here. Do you refer to acclimation of the pine tree density or leaf area to the climate at Yatir? Or to the general adaptation of P. halepensis as a species, to dry climates?

A: We have elaborated the discussion section. With the adaptation of the tree species we are referring to the effective counter measures such as a very early onset of photosynthesis downregulation or the high resistance of vulnerability as expressed in the PLC curve (L509ff)

L476-479 – Note that soil-to-root conductance can strongly decrease but still have your plants connected to the soil. In addition to reduction of conductance (or disconnection, as in your case), one needs explicit (or implicit) water compartments to achieve plant water potentials less negative than soil water potentials in a model, regardless of the complexity of hydraulics.

A: We are now noting the consideration of water compartments in addition to the claim for simplification (L550ff).

L496 – Here you could mention other sites (e.g. Puechabon, EucFACE) where litterfall can be more safely attributed to drought and, therefore, would be more amenable for testing the importance of simulating drought-related leaf senescence.

A: We are now mentioning that an elaborated evaluation requires model applications at other long-term observations sites in dry regions [e.g. in France or Italy (Reichstein et al., 2002)] (L572/573).

L529 – In SurEau, gmin is dependent on Tleaf
(https://gmd.copernicus.org/articles/15/5593/2022/)

A: It is now mentioned that GMIN in the SurEau model is temperature dependent (L606/7).

Mentioned references:

Gourlez de la Motte, L., Beauclaire, Q., Heinesch, B., Cuntz, M., Foltýnová, L., Sigut, L., Manca, G., Ballarin, I., Vincke, C., Roland, M., Ibrom, A., Lousteau, D., and Bernard, L.: Non-stomatal processes reduce gross primary productivity in temperate forest ecosystems during severe edaphic drought, Philosophical Transactions of the Royal Society B: Biological Sciences, 375, 20190527, 10.1098/RSTB-2019-0527, 2020.

Keenan, T., Sabaté, S., and Gracia, C.: Soil water stress and coupled photosynthesis-conductance models: Bridging the gap between conflicting reports on the relative roles of stomatal, mesophyll conductance and biochemical limitations to photosynthesis, Agric. Forest Meteorol., 150, 443-453, 2010.

Kennedy, D., Swenson, S., Oleson, K. W., Lawrence, D. M., Fisher, R., Lola da Costa, A. C., and Gentine, P.: Implementing Plant Hydraulics in the Community Land Model, Version 5, J. Adv. Model. Earth Syst., 11, 485-513, 10.1029/2018MS001500, 2019.

Kuusk, V., Niinemets, Ü., and Valladares, F.: Structural controls on photosynthetic capacity through juvenile-to-adult transition and needle ageing in Mediterranean pines, Funct. Ecol., 32, 1479-1491, 10.1111/1365-2435.13087, 2018.

Reichstein, M., Tenhunen, J. D., Roupsard, O., Ourcival, J.-M., Rambal, S., Dore, S., and Valentini, R.: Ecosystem respiration in two Mediterranean evergreen Holm Oak forests: drought effects and decomposition dynamics, Funct. Ecol., 16, 27-39, 10.1046/j.0269-8463.2001.00597.x, 2002.

Rodriguez-Dominguez, C. M., and Brodribb, T. J.: Declining root water transport drives stomatal closure in olive under moderate water stress, New Phytol., 225, 126-134, 10.1111/nph.16177, 2020.

Yang, J., Duursma, R. A., De Kauwe, M. G., Kumarathunge, D., Jiang, M., Mahmud, K., Gimeno, T. E., Crous, K. Y., Ellsworth, D. S., Peters, J., Choat, B., Eamus, D., and Medlyn, B. E.: Incorporating non-stomatal limitation improves the performance of leaf and canopy

models at high vapour pressure deficit, Tree Physiol., 39, 1961–1974, 10.1093/treephys/tpz103, 2019.

Yao, Y., Joetzjer, E., Ciais, P., Viovy, N., Cresto Aleina, F., Chave, J., Sack, L., Bartlett, M., Meir, P., Fisher, R., and Luyssaert, S.: Forest fluxes and mortality response to drought: model description (ORCHIDEE-CAN-NHA, r7236) and evaluation at the Caxiuanã drought experiment, Geosci. Model Dev., 15, 7809–7833, 10.5194/gmd-15-7809-2022, 2022.

---

## Author Response (AR2)

Comments to the general suggestions and questions (from reviewer 2)

R2: I found the manuscript overall improved with respect to the previous version. The authors have successfully addressed my previous concern regarding the statement of objectives and tried to illustrate the effects of considering the different processes. The new model is overall easier to understand now, but the section on Psi_dehydration and how plant draws from water storage for transpiration after soil disconnection is still hard to understand.

A: Thanks for the acknowledgement of the improvements. We are happy that the new version is clearer and easier to understand now, which is also due to the good comments. Regarding the description of water uptake and dehydration, we have tried to further improve it (L282-297, see also specific comment below).

R2: I wonder whether the need to include NSL for proper stomatal behavior (Fig. 5) is related to the fact that direct hydraulic effects on stomatal conductance are based on a rather resistant stem vulnerability curve. Despite the better fit of transpiration when including NSL, it would be interesting to see whether the fit GPP also improves.

A: Indeed, a resistant vulnerability curve and a strong dependence of stomatal conductance on NSL seems to be a likely trait combination under dry conditions since it enables a high drought stress resistance. Despite this is clearly supported by our study as well as the measurements at the site, it is still not quite clear from literature results. We have added this remark in the discussion (L523-25).

The modeled GPP response depends indeed strongly on the NSL effect. In our simulations, the NSL effect is determined via our initial Bayesian parameter calibration to GPP. In fact, the sensitivity of the model to NSL seems quite large, as indicated in the example figure below were PSI_EXP has been set to zero, letting all the other parameters untouched. It should be noted that the (overall) 28 % higher GPP (difference between the trend lines in the figure) in the simulations goes along with an overestimation of evaporation and a consequently unrealistic drop of predawn water potential down to -4MPa (not shown), which has been also pointed out by Sabot et al. (2022).

Sabot, M. E., De Kauwe, M. G., Pitman, A. J., Medlyn, B. E., Ellsworth, D. S., Martin-StPaul, N. K., ... & Serbin, S. P. (2022). One stomatal model to rule them all? Toward improved representation of carbon and water exchange in global models. *Journal of Advances in Modeling Earth Systems*, *14*(4), e2021MS002761.

[Figure]

Comments to the specific suggestions and questions (from reviewer 2)

L34: "it" also disclosed

A: corrected

L299: How is the value of soil water uptake (UPTsw) determined?

A: UPTsw is defined by transpiration demand as long as the soil water potential threshold is not reached. Across the soil profile, water in different layers is taken according to their relative water content as well as fine root abundance. We have added these explanations in L282-285.

L607. Package MEDFATE (4.1.0) implements SurEau, including capacitance effects and temperature-effects on gmin.

A: MEDFATE has been mentioned, capacitance has been added and the reference has been switched to De Cáceres et al. (L609-611)

De Cáceres, M., Molowny-Horas, R., Cabon, A., Martínez-Vilalta, J., Mencuccini, M., García-Valdés, R., Nadal-Sala, D., Sabaté, S., Martin-StPaul, N., Morin, X., D'Adamo, F., Batllori, E., & Améztegui, A. (2023). MEDFATE 2.9.3: a trait-enabled model to simulate Mediterranean forest function and dynamics at regional scales. *Geoscientific Model Development*, *16*(11), 3165-3201. https://doi.org/10.5194/gmd-16-3165-2023

In addition, we like to indicate that we have switched the order of the first two paragraphs in the discussion and homogenized the different naming of krc/kxyl in figure and caption of Fig. 1.